# Single amino acid residue mediates reciprocal specificity in two mosquito odorant receptors

**Flavia P Franco[1], Pingxi Xu[1], Brandon J Harris[2], Vladimir Yarov-Yarovoy[2,3], Walter S Leal[1]***

[1]Department of Molecular and Cellular Biology, University of California, Davis, Davis, United States; [2]Department of Physiology and Membrane Biology, University of California, Davis, Davis, United States; [3]Department of Anesthesiology and Pain Medicine, University of California, Davis, Davis, United States

**Abstract** The southern house mosquito, *Culex quinquefasciatus,* utilizes two odorant receptors, CquiOR10 and CquiOR2, narrowly tuned to oviposition attractants and well conserved among mosquito species. They detect skatole and indole, respectively, with reciprocal specificity. We swapped the transmembrane (TM) domains of CquiOR10 and CquiOR2 and identified TM2 as a specificity determinant. With additional mutations, we showed that CquiOR10A73L behaved like CquiOR2. Conversely, CquiOR2L74A recapitulated CquiOR10 specificity. Next, we generated structural models of CquiOR10 and CquiOR10A73L using RoseTTAFold and AlphaFold and docked skatole and indole using RosettaLigand. These modeling studies suggested space-filling constraints around A73. Consistent with this hypothesis, CquiOR10 mutants with a bulkier residue (Ile, Val) were insensitive to skatole and indole, whereas CquiOR10A73G retained the specificity to skatole and showed a more robust response than the wildtype receptor CquiOR10. On the other hand, Leu to Gly mutation of the indole receptor CquiOR2 reverted the specificity to skatole. Lastly, CquiOR10A73L, CquiOR2, and CquiOR2L74I were insensitive to 3-ethylindole, whereas CquiOR2L74A and CquiOR2L74G gained activity. Additionally, CquiOR10A73G gave more robust responses to 3-ethylindole than CquiOR10. Thus, we suggest the specificity of these receptors is mediated by a single amino acid substitution, leading to finely tuned volumetric space to accommodate specific oviposition attractants.

**\*For correspondence:**
wsleal@ucdavis.edu

**Competing interest:** The authors declare that no competing interests exist.

## Editor's evaluation

This article addresses the mechanism of ligand specificity of odorant receptors (OR) through mutational analyses and structure prediction. Through solid data, the authors identify a single amino acid substitution that switches ligand specificity between two olfactory receptors. Obtaining structures of OR complexes has been challenging, so such an approach is valuable and will be of interest to scientists within the fields of chemical ecology and sensory neuroscience.

## Introduction

Insects perceive the world with a sophisticated olfactory system essential for survival and reproduction. Their antennae are biosensors *par excellence,* allowing detection of a plethora of compounds, some with extraordinary sensitivity and selectivity (*Kaissling, 2014*). The insect olfactory system is comprised mainly of odorant-binding proteins, odorant-degrading enzymes, ionotropic receptors, and the ultimate gatekeepers of selectivity (*Leal, 2013*; *Leal, 2016*; *Leal, 2020*) – the odorant

receptors (ORs) (*Clyne et al., 1999*; *Gao and Chess, 1999*; *Vosshall et al., 1999*). The ORs are the binding units in functional heteromeric cation channels (*Neuhaus et al., 2005*) formed with an odorant receptor coreceptor (Orco) (*Larsson et al., 2004*). Unlike mammalian olfactory receptors, insect ORs and Orco have inverse topologies compared to G-protein-coupled receptors (GPCRs), with a cytosolic N-terminus and an extracellular C-terminus (*Benton et al., 2006*; *Lundin et al., 2007*). One of the major breakthroughs in the fields of insect olfaction in the last two decades since the discovery of ORs (*Leal, 2020*) was the determination of the cryo-electron microscopy (cryo-EM) structure of the Orco homomer from the parasitic fig wasp, *Apocrypta bakeri,* AbakOrco (*Butterwick et al., 2018*). Subsequently, the structure for a promiscuous OR from the evolutionarily primitive (Apterygota, wingless) jumping bristletail, *Machilis hrabei,* MhraOR5, was solved (*DelMarmol et al., 2021*). Although structures of ORs from winged insects (Pterygota) have not been solved to date, amino acid residues critical for OR specificity have been reported (*Auer et al., 2020*; *Cao et al., 2021*; *Hughes et al., 2014*; *Leary et al., 2012*; *Pellegrino et al., 2011*; *Yang et al., 2017*; *Yuvaraj et al., 2021*). These studies focused on one-way alteration of specificity but did not examine how an insect detects two odorants with reverse specificity.

Mosquitoes are vectors of pathogens that cause tremendous harm to public health. Male and female mosquitoes visit plants to obtain nutrients for flight. For reproduction and survival of the species, females must acquire a blood meal to fertilize their eggs and, subsequently, oviposit in an aquatic environment suitable for the offspring to flourish. While feeding on hosts, females transmit viruses and other pathogens. The southern house mosquito, *Culex quinquefasciatus,* transmits pathogens causing filariasis and various encephalitis (*Nasci and Miller, 1996*). In the United States, mosquitoes belonging to the *Culex pipiens* complex transmit the West Nile virus (*Andreadis, 2012*). Due to its opportunistic feeding on avian and mammalian hosts, *Cx. quinquefasciatus* is a significant bridge vector in urbanized centers in the Western United States, particularly southern California (*Andreadis, 2012*; *Syed and Leal, 2009*). Female mosquitoes rely on multiple sensory modalities, including olfaction, to find plants, vertebrate hosts, and suitable environments for oviposition.

The genome of the southern house mosquito, *Cx. quinquefasciatus* (*Arensburger et al., 2010*), has the most extensive repertoire of *OR* genes (*Leal et al., 2013*) among mosquito species. The genomes of *Anopheles darlingi,* the malaria mosquito, *Anopheles gambiae,* the yellow fever mosquito, *Aedes aegypti*, the Asian tiger mosquito, *Aedes albopictus,* and the southern house mosquito contain 18 (*Marinotti et al., 2013*), 79 (*Hill et al., 2002*), 117–131 (*Bohbot et al., 2007*; *Nene et al., 2007*), 158 (*Chen et al., 2015*), and 180 (*Arensburger et al., 2010*) *OR* genes, respectively. Of those, 61 transcripts were found in *An. gambiae* (*Pitts et al., 2011*), 107 in *Ae. aegypti* (*Matthews et al., 2018*), and 177 in *Cx. quinquefasciatus* (*Leal et al., 2013*). Given that the number of odorants in the environment is larger than the number of OR genes even in *Cx. quinquefasciatus*, it is not surprising that many ORs from insects are promiscuous (*Leal, 2013*; *Pask, 2020*). However, ORs detecting behaviorally critical compounds (semiochemicals) may be narrowly tuned (*Hughes et al., 2010*; *Nakagawa et al., 2005*; *Stensmyr et al., 2012*).

ORs narrowly tuned to 3-methylindole (=skatole) and indole have been found in mosquito species in the subfamilies Culicinae and Anophelinae. Specifically, OR10 and OR2 have been de-orphanized in the southern house mosquito (*Hughes et al., 2010*; *Pelletier et al., 2010*), the yellow fever mosquito (*Bohbot et al., 2011*), and the malaria mosquito (*Carey et al., 2010*; *Wang et al., 2010*). Recently, these so-called indolergic receptors *Bohbot and Pitts, 2015* have also been found in the housefly, *Musca domestica* (*Pitts et al., 2021*). Skatole and indole are fecal products that have been identified as oviposition attractants for the southern house mosquito (*Blackwell et al., 1993*; *Mboera et al., 2000*; *Millar et al., 1992*). Skatole is a potent oviposition attractant in minute doses, whereas indole is active only at high doses (*Millar et al., 1992*). To the human nose, skatole has a pungent fecal odor. In contrast, indole has an almost floral odor when highly purified and presented at low doses (*Fenaroli, 1975*), but a fecal odor at high doses. Remarkably, in all mosquito species and the housefly, OR10s are narrowly tuned to skatole and respond with lower sensitivity to indole, whereas indole is the most potent ligand for OR2s, which give lower responses to skatole. In *Cx. quinquefasciatus,* these receptor genes were initially named *CquiOR10* and *CquiOR2*, respectively, renamed *CquiOR21* and *CquiOR121*, but a proposition to restore the original names is under consideration (Carolyn McBride, personal communication). Here, we refer to these receptors from *Cx. quinquefasciatus* as CquiOR10

(VectorBase and GenBank IDs, CPIJ002479 and GU945397, respectively) and CquiOR2 (CPIJ014392, GU945396.1).

These oviposition attractant-detecting (*Chen and Luetje, 2014*) receptors in *Cx. quinquefasciatus*, CquiOR10 and CquiOR2, provide a suitable model to identify specific determinants as they respond to skatole and indole with reverse specificity. CquiOR10 and CquiOR2 proteins have 377 and 375 amino acid residues, respectively, and share only 49.5% amino acid identity, whereas 61.9% of the amino acids in the predicted transmembrane domains are identical (*Supplementary file 1*, Table 1). To identify the specificity determinants of these receptors, we swapped transmembrane (TM) domains and tested the chimeric receptors using the *Xenopus* oocyte recording system. Given that CquiOR10 is the second most sensitive *Cx. quinquefasciatus* OR (second only to CquiOR36; *Choo et al., 2018*), we replaced the predicted (*Reynolds et al., 2008*) transmembrane domains from CquiOR2 into CquiOR10 and measured the specificity of the chimeric receptors. With this approach, we identified TM2 as a specificity determinant. Next, we tested mutations of chimeric CquiOR10 and identified a single amino acid residue in TM2 (Ala-73) that determines the receptor's specificity. We then directly mutated the wildtype receptor and observed that CquiOR10A73L is specific to indole as CquiOR2. Additionally, CquiOR2L74A emulated the response profile of CquiOR10. To better understand the structural basis of this single-point specificity determinant, we generated structural models of CquiOR10 and CquiOR10A73L using RoseTTAFold (*Baek et al., 2021*) and AlphaFold (*Jumper et al., 2021*). We identified the binding poses for skatole and indole using RosettaLigand molecular docking and observed a finely tuned volumetric space to accommodate specific oviposition attractants.

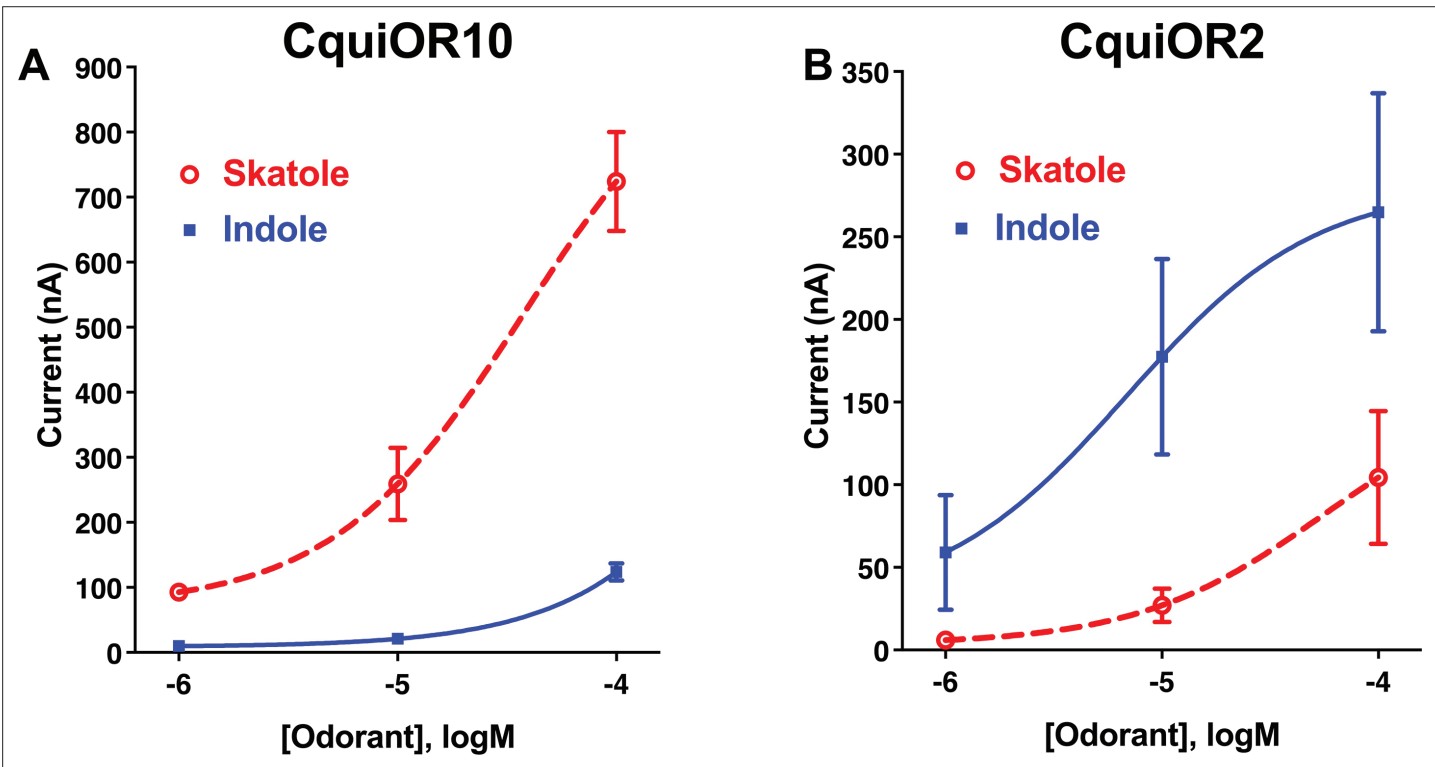

**Figure 1.** Concentration–response analysis for activation of wildtype odorant receptors (ORs) by skatole and indole. (**A**) CquiOR10 and (**B**) CquiOR2. Lines were obtained with nonlinear fit. Bars represent SEM. n = 4–5.

The online version of this article includes the following source data and figure supplement(s) for figure 1:

**Source data 1.** Concentration–response analysis for activation of wildtype odorant receptors (ORs) by skatole and indole.

**Figure supplement 1.** Concentration–response analysis for wildtype and chimeric odorant receptors (ORs).

# Results

## Chimeric OR with reversed specificity

We envisioned that studying a pair of ORs with reverse specificities, like CquiOR10 and CquiOR2, could lead us to specificity determinants. CquiOR10 is activated by the oviposition attractant skatole (*Blackwell et al., 1993*; *Mboera et al., 2000*; *Millar et al., 1992*) with high specificity (*Figure 1A*), whereas CquiOR2 is specific to indole (*Figure 1B*).

Our approach was designed to swap TM domains using the more sensitive receptor, CquiOR10, as the acceptor. Specifically, we generated chimeric receptors by replacing CquiOR10 TM domains with related domains from CquiOR2 (*Figure 2A*). During the life of this project, the cryo-EM structure of an odorant receptor coreceptor AbakOrco from the parasitic fig wasp, *Aprocrypta bakeri*, was reported (*Butterwick et al., 2018*). We then compared the experimental structure (*Butterwick et al., 2018*) with the predicted topology for AbakOrco using the same OCTOPUS method (*Viklund and Elofsson, 2008*) we used to identify CquiOR10 and CquiOR2 TMs (*Figure 2A*; *Viklund and Elofsson, 2008*). The almost perfect overlap between OCTOPUS prediction and the AbakOrco structure (*Figure 2B*) validated not only our TM predictions (*Figure 2A*), but also the 21 chimeric ORs already tested when the structure of the coreceptor AbakOrco (*Butterwick et al., 2018*) was reported.

We referred to these chimeric receptors as CquiOR10$^{Mx}$, where Mx refers to TMx from CquiOR2. We performed functional assays of these CquiOR10$^{Mx}$ receptors using the *Xenopus* oocyte recording system. This long-term project could require as many as 127 possible chimeric receptors. We started by swapping all seven TM domains. We envisioned that this chimeric receptor would have a reverse specificity. If so, we would restore one TM at a time to identify critical domains. It turned out that CquiOR10$^{M1,2,3,4,5,6,7}$ was silent (see Appendix 1, *Supplementary file 1*, Table 2). To minimize the number of tested mutants, we changed the strategy to start from single mutations to obtain educated guess for the subsequent design of mutants. With this approach, we generated and tested only 8 of the required 99 mutants with 7–3 TMs swapped. We tested 36 chimeric receptors (see *Supplementary file 1*, Table 2, and *Figure 1—figure supplement 1*). Fourteen chimeric receptors did not respond to skatole or indole, and 21 receptors retained the specificity to skatole (*Supplementary file 1*, Table 2, and *Figure 1—figure supplement 1*). Lastly, CquiOR10$^{M2,7}$/CquiOrco-expressing oocytes responded to both skatole and indole with a reverse profile (*Figure 3A*). This dataset shows that CquiOR10$^{M2,7}$ emulated the profile of the indole receptor CquiOR2 (*Figure 1B*).

## A single-point mutation that reverses the specificity of the skatole and indole receptors

We proceeded to identify the amino acid residues in the swapped domains of CquiOR10$^{M2,7}$, directly affecting the specificity of the chimeric and wildtype receptors. Given the observation that, by and large, chimeric receptors with TM5 and TM6 from CquiOR2 gave stronger responses (*Figure 1—figure supplement 1*), we asked whether CquiOR10$^{M2,7}$ responses with these two additional TM domains swapped would give more robust responses while keeping the same specificity to indole. CquiOR10$^{M2,5,6,7}$/CquiOrco-expressing oocytes were indeed more sensitive while maintaining the selectivity to indole (*Figure 3B*). We then used CquiOR10$^{M2,5,6,7}$ and designed various mutants to rescue single or multiple residues in TM2 at a time (*Figure 4*). We focused on TM2 because swapping TM7 did not affect the specificity of the receptor (*Figure 1—figure supplement 1E*). We divided TM2 into outer, middle, and inner segments based on the topology predicted by OCTOPUS (*Viklund and Elofsson, 2008*).

It has been postulated that the extracellular halves of TM domains form an odorant-binding pocket (*Guo and Kim, 2010*); thus, we first examined a mutant (CquiOR10$^{M2,5,6,7}$_Outer) having the residues in the outer segment at 59, 60, 63, and 64 restored to Glu, Val, Asn, and Ala, respectively, as in the wildtype receptor (*Figure 4*). CquiOR10$^{M2,5,6,7}$_Outer/CquiOrco-expressing oocytes retained the specificity of CquiOR10$^{M2,5,6,7}$ with a more robust response to indole than skatole (*Figure 3C*). These findings indicated that the residues in the outer segment of TM2 are not specificity determinants. After that, we tested a chimeric receptor with the residues in the middle and inner part of the TM2 domain rescued to match those in the wildtype receptor (*Figure 4*). CquiOR10$^{M2,5,6,7}$_Mid;Inner restored the skatole-specific profile of CquiOR10 (*Figure 3D*), thus suggesting that amino acid residues in these segments are specificity determinants. Then, we tested CquiOR10$^{M2,5,6,7}$_Inner (=CquiOR10$^{M2,5,6,7}$L73A

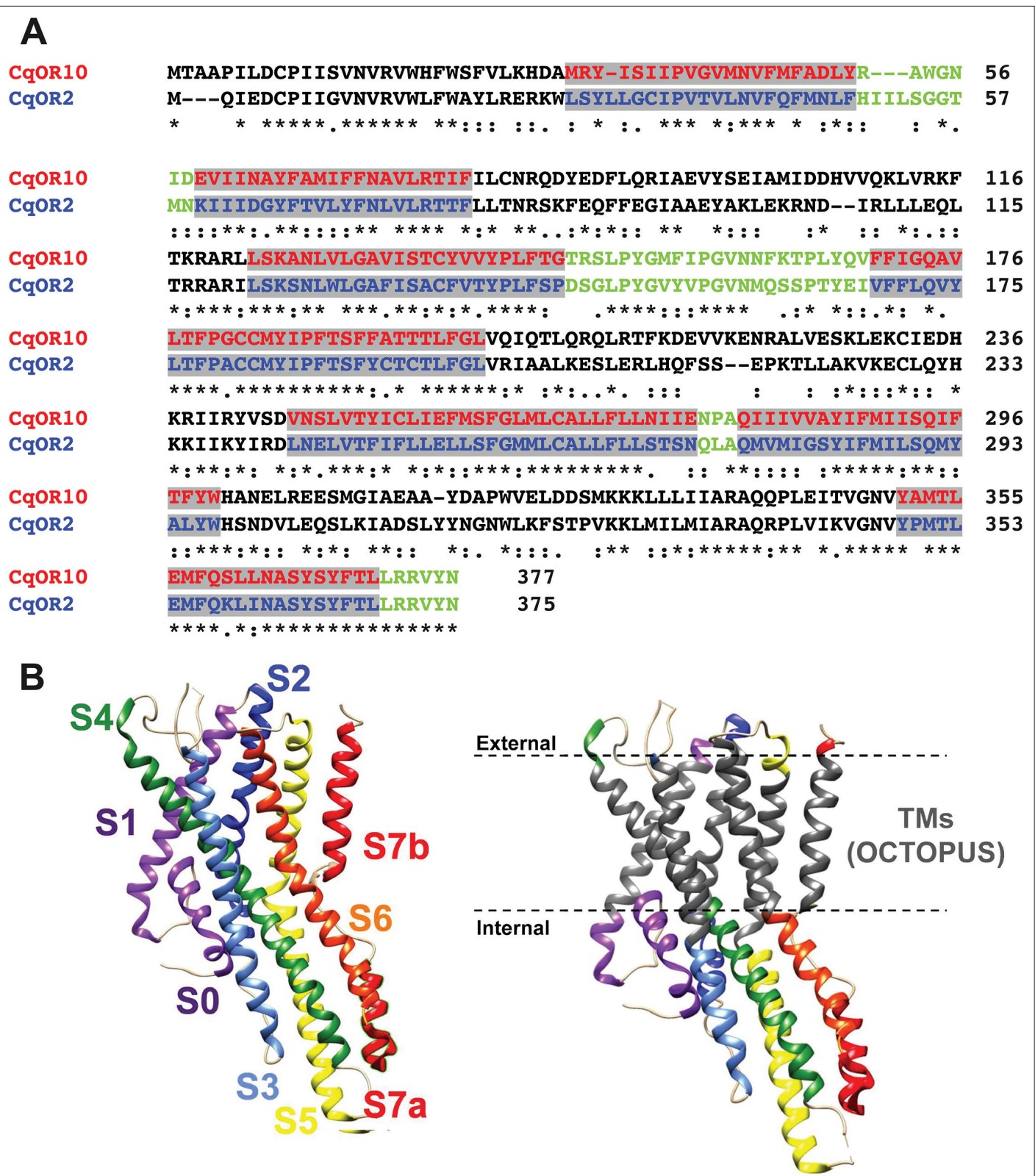

**Figure 2.** Alignment of the amino acid sequences of CquiOR10 and CquiOR2 highlighting the predicted transmembrane (TM) domains and a comparison of predicted and experimentally determined TM domains of the odorant receptor coreceptor, AbakOrco. (**A**) CqOR10 and CqOR2 are abbreviations for CquiOR10 and CquiOR2, respectively. The TM domains, predicted by OCTOPUS, are displayed in red and blue for CquiOR10 and CquiOR2, respectively. The sequences of the N-terminus and the intracellular loops are displayed in black, and the C-terminus and extracellular loops

*Figure 2 continued on next page*

*Figure 2 continued*

in green. (**B**) Left: the cryo-EM structure of AbakOrco (PDB, 6C70) displayed in rainbow color using UCSF Chimera (*Pettersen et al., 2004*). Right: the predicted TM domains (right) are displayed in gray. The dashed lines represent the membrane boundaries.

;T78I), which had only residues at 73 and 78 rescued to the wildtype Ala and Ile, respectively. Oocytes co-expressing CquiOR10$^{M2,5,6,7}$_Inner and CquiOrco reverted the specificity to skatole (*Figure 3E*). As a result, we concluded that residues in the predicted inner part of TM2 are critical for the chimeric receptor's specificity. We further probed the chimeric receptor CquiOR10$^{M2,5,6,7}$ with single-point mutations to identify the residue(s) determining specificity. CquiOR10$^{M2,5,6,7}$T78I/CquiOrco-expressing oocytes showed the same specificity as the chimeric receptor CquiOR10$^{M2,5,6,7}$ (*Figure 3F*). Specifically, CquiOR10$^{M2,5,6,7}$T78I gave a more robust response to indole than skatole, suggesting that rescuing the residue at 78 did not affect CquiOR10$^{M2,5,6,7}$ specificity. By contrast, CquiOR10$^{M2,5,6,7}$L73A/CquiOrco-expressing oocytes reverted the specificity to skatole (*Figure 3G*), thus behaving like the wildtype receptor CquiOR10 (*Figure 1A*). To further examine the role of Ala-73 as a specificity determinant residue, we obtained a single-point mutation of CquiOR10$^{M7}$, which is specific to skatole (*Figure 1—figure supplement 1E*). The responses recorded from CquiOR10$^{M7}$A73L/CquiOrco-expressing oocytes showed a reverse, indole-specific profile (*Figure 3H*), like the CquiOR2 profile (*Figure 1B*). Having identified a single amino acid residue in the chimeric receptor that switches the skatole/indole specificity, we tested the effect of single-point mutation on the specificity of the wildtype receptor CquiOR10 (EC50: skatole, 3.6 µM; indole 29.9 µM). CquiOR10A73L showed a reverse specificity, with dose-dependent responses to indole (*Figure 3I*) (EC50: indole, 3.4 µM; skatole 53.7 µM). Collectively, these findings suggest that a single amino acid residue in CquiOR10 determines the specificity of this receptor. Additionally, we obtained an equivalent single-point mutation in the indole-specific CquiOR2 (*Figure 1B*) (EC50: indole, 7.7 µM; skatole 16.4 µM). Thus, CquiOR2L74A/CquiOrco-expressing oocytes gave robust and specific responses to skatole (*Figure 3J*) (EC50: skatole, 8.5 µM; indole 27.6 µM).

As summarized in a graphical representation (*Figure 3—figure supplement 1*), these findings demonstrate that these two mosquito odorant receptors, CquiOR10 and CquiOR2, have reciprocal specificity mediated by a single amino acid residue, Ala-73 and Leu-74, respectively.

We also recorded the response of these ORs to other phenolic ligands that activate indolic receptors, albeit generating small currents. While CquiOR10 and CquiOR2 responded to phenol, 3,5-dimethylphenol activated only CquiOR10 (*Figure 5A*). A single-point mutation in CquiOR10 rendered the chimeric receptor insensitive to 3,5-dimethylphenol. By contrast, an equivalent mutation in CquiOR2 recapitulated the profile of CquiOR10 (*Figure 5A*).

Additionally, we recorded responses elicited by methylindoles. Specifically, we challenged oocytes with 1-methylindole, 2-methylindole, 4-methylindole, 5-methylindole, 6-methylindole, and 7-methylindole. In these analyses, we did not stimulate the oocyte preparations with 3-methylindole to avoid possible desensitization. CquiOR10/CquiOrco-expressing oocytes elicited stronger responses when challenged with 1-methylindole and 5-methylindole than when stimulated with the other methylindoles (*Figure 5B*). By contrast, CquiOR2/CquiOrco-expressing oocytes elicited similarly lower responses when stimulated with methylindoles (*Figure 5C*). CquiOR2L74A with a single-point mutation to mimic OR10 receptor recapitulated CquiOR10 response profile (*Figure 5D*). These data suggest that a single amino acid residue determines a receptor's specificity toward ligands eliciting robust or small responses.

## CquiOR10 computational modeling suggests space-filling constraints for indole-based odorants around A73

To structurally hypothesize the above-described reciprocal specificity, we generated structural models of CquiOr10, CquiOR2, CquiOR10A73L, and CquiOR2L74A using RoseTTAFold (*Baek et al., 2021*) and a structural model of CquiOR10 using AlphaFold (*Jumper et al., 2021*; *Figure 6*).

CquiOR10 models of one homotetramer subunit were generated with AlphaFold and RoseTTA-Fold, each producing five models. RoseTTAFold models of CquiOR10, CquiOR10A73L, CquiOR2, and CquiOR2A73L produced a transmembrane helix root mean square deviation (RMSD) between α-Carbon atoms less than 1 Å across all 20 models (five models per odorant receptor); this suggests that the homologous CquiOR10 and CquiOR2 are structurally similar and that single-point mutations

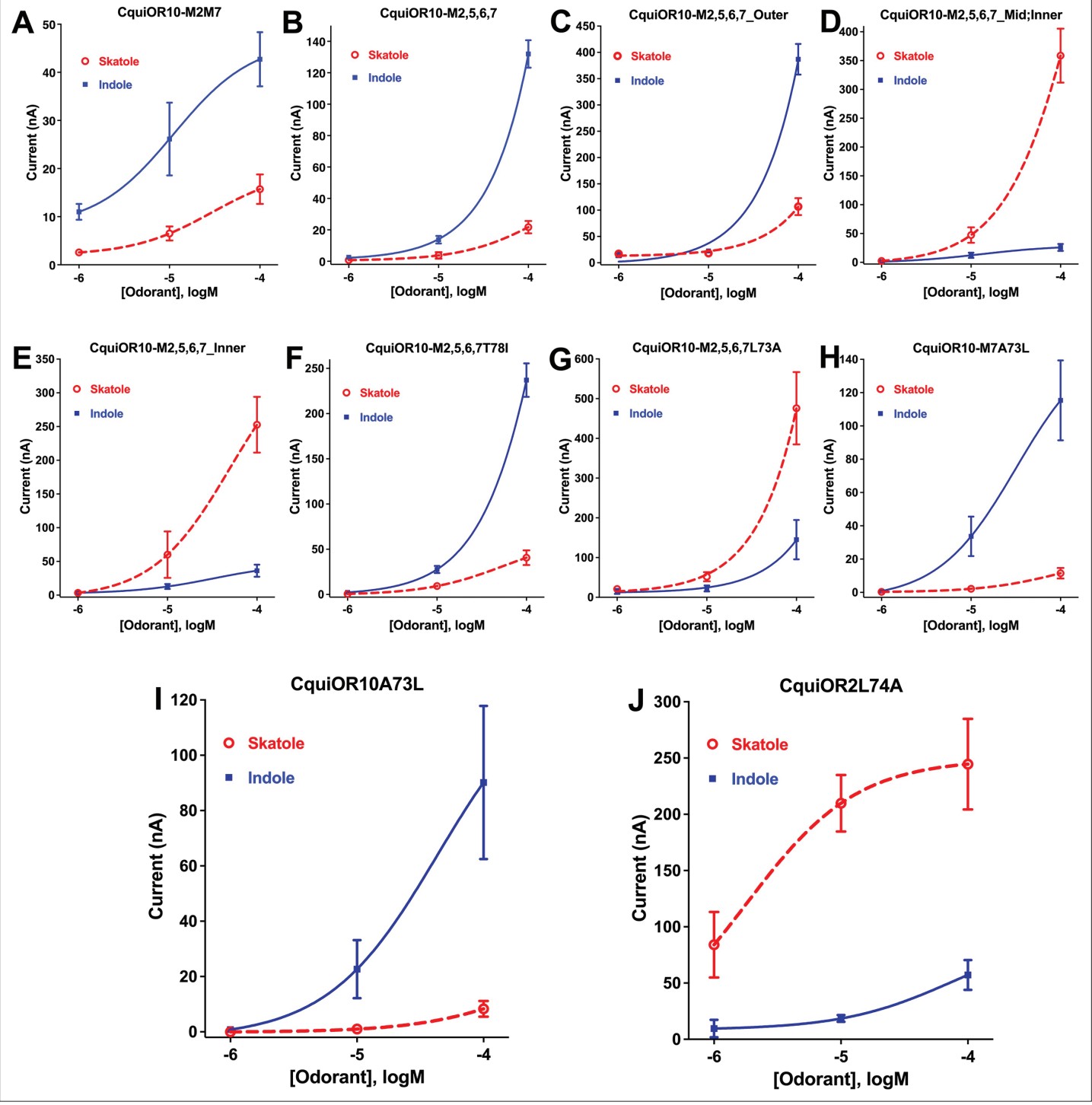

**Figure 3.** Concentration–response curves obtained with chimeric odorant receptors (ORs) stimulated with skatole and indole. (**A**) CquiOR10$^{M2,7}$;
(**B**) CquiOR10$^{M2,5,6,7}$; (**C**) CquiOR10$^{M2,5,6,7}$_Outer; (**D**) CquiOR10$^{M2,5,6,7}$_Mid;Inner; (**E**) CquiOR10$^{M2,5,6,7}$_Inner; (**F**) CquiOR10$^{M2,5,6,7}$T78I; (**G**) CquiOR10$^{M2,5,6,7}$L73A;
(**H**) CquiOR10$^{M7}$A73L; (**I**) CquiOR10A73L; (**J**) CquiOR2L74A. Lines were obtained with nonlinear fit. Bars represent SEM. The number of replicates (n) were
7, 4, 5, 5, 4, 3, 9, 7, 6, and 5, respectively.

The online version of this article includes the following source data and figure supplement(s) for figure 3:

**Source data 1.** Concentration–response curves obtained with chimeric odorant receptors (ORs) stimulated with skatole and indole.

**Figure supplement 1.** Schematic view of the workflow.

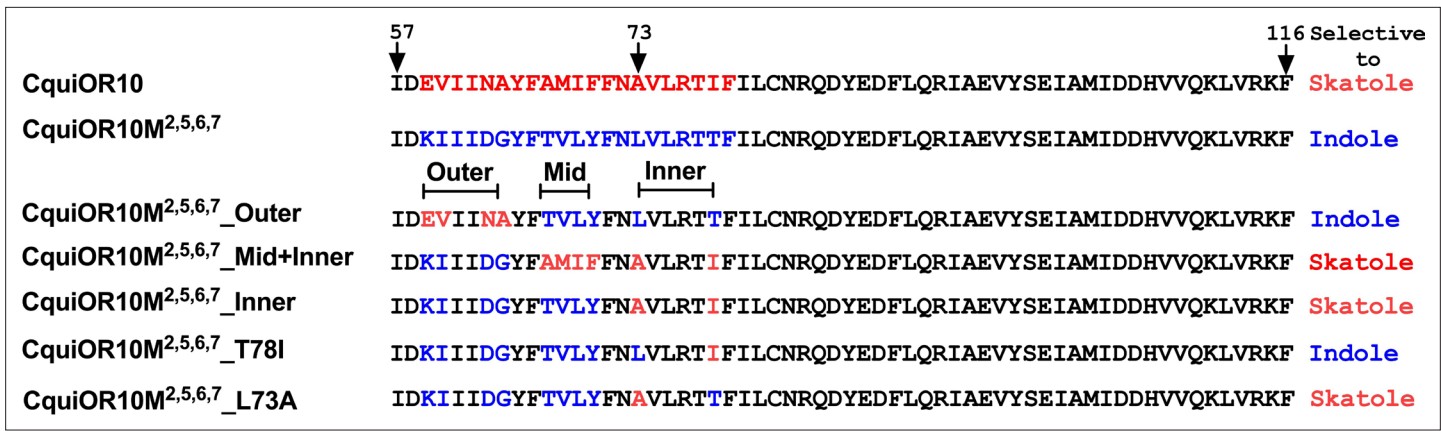

**Figure 4.** Partial sequences of CquiOR10 and chimeric odorant receptors (ORs) highlighting transmembrane domain-2 (TM2). The two last residues of the extracellular loop-1 (Ile-57 and Asp-58) appear in the N-terminus. The TM2 was divided into the arbitrary segments outer, middle (mid), and inner to identify specificity determinants.

should not cause great structural deviation from wildtype. Comparing CquiOR10 models, RoseTTAFold and AlphaFold produced a transmembrane helix RMSD of 1.7 Å, with the transmembrane helices at the extracellular membrane face having the largest structural deviation. Considering there is little structural knowledge of insect odorant receptors, their binding mechanisms, and their conformational changes, we suspected that homologous odorant receptors would have similar binding modes. Further, pairwise sequence alignment suggests that a series of residues in MhraOR5 TM4 aligns with CquiOR10 TM2, which contains CquiOR10A73 (CquiOR10: [59]EVI-INAYFAMIFFNAV[74]. MhraOR5: [199]EVIAIYEAVAMIFLITA[215.]; ***Figure 6—figure supplement 1***) while AlphaFold and RoseTTAFold models of CquiOR10 were broadly similar to MhraOR5 (***Figure 6—figure supplement 2***). Using transmembrane helix 7b (TM7b), we superimposed the top-ranking CquiOR10 RoseTTAFold and AlphaFold models with an experimentally resolved structure, *M. hrabei* (MhraOR5) in complex with eugenol (PDB ID: 7LID; ***DelMarmol et al., 2021***) to identify which of our models resembled an odorant-bound conformation. With this selection criteria, we proceeded with RoseTTAFold models of the odorant receptors for Rosetta-based small-molecule docking method RosettaLigand (***Davis and Baker, 2009***; ***DeLuca et al., 2015***) as the structural similarity around the hypothesized binding pocket was greater than the AlphaFold models of the odorant receptors compared with the MhraOR5 structure. We chose to select conformationally similar models over modeling and docking an apo structure into a bound conformation because it is a more cautious approach when there is little structural information. We perceived modeling an apo structure into a bound conformation to potentially yield more biologically implausible conformations than docking of a structurally comparative model.

To verify that RosettaLigand could effectively sample odorants in receptors homologous to CquiOR10 and CquiOR2, we used the structure of eugenol in complex with the insect odorant receptor OR5 from MhraOR5 (PDB ID: 7LID) as a control (***DelMarmol et al., 2021***). Rosetta protein-ligand docking employs energy-based analyses, such as the interface energy between protein and ligand, to select the representative models (***Bidula et al., 2022***). With this selection method, the RMSD of our MhraOR5-eugenol models relative to the experimental structure ranged from 0.5 to 5.0 Å (***Supplementary file 1***, Table 3). RMSD values equal or below 2.0 Å are considered an appropriate range for validation (***Park et al., 2021***). After using hdbscan cluster analysis (***McInnes et al., 2017***) to group structurally similar models, the largest cluster had a RMSD of 0.75 Å while the lowest interface-energy model had a RMSD of 2.4 Å. Collectively, these data demonstrate that RosettaLigand paired with the hdbscan clustering method can recapitulate the MhraOR5 structure and blindly select a near-native model, thus is suitable for structural predictions of odorants with CquiOR10A73L and CquiOR2L74A (***Figure 6—figure supplements 3 and 4***).

For each receptor–ligand complex (CquiOR10-skatole, CquiOR10A73L-skatole, CquiOR10-indole, and CquiOR10A73L-indole), we generated 100,000 docking models using RosettaLigand, clustered the 10,000 lowest interface-energy models, and selected the lowest interface-energy model from the 10 largest clusters, resulting in 10 models per receptor–ligand complex from which to draw structural

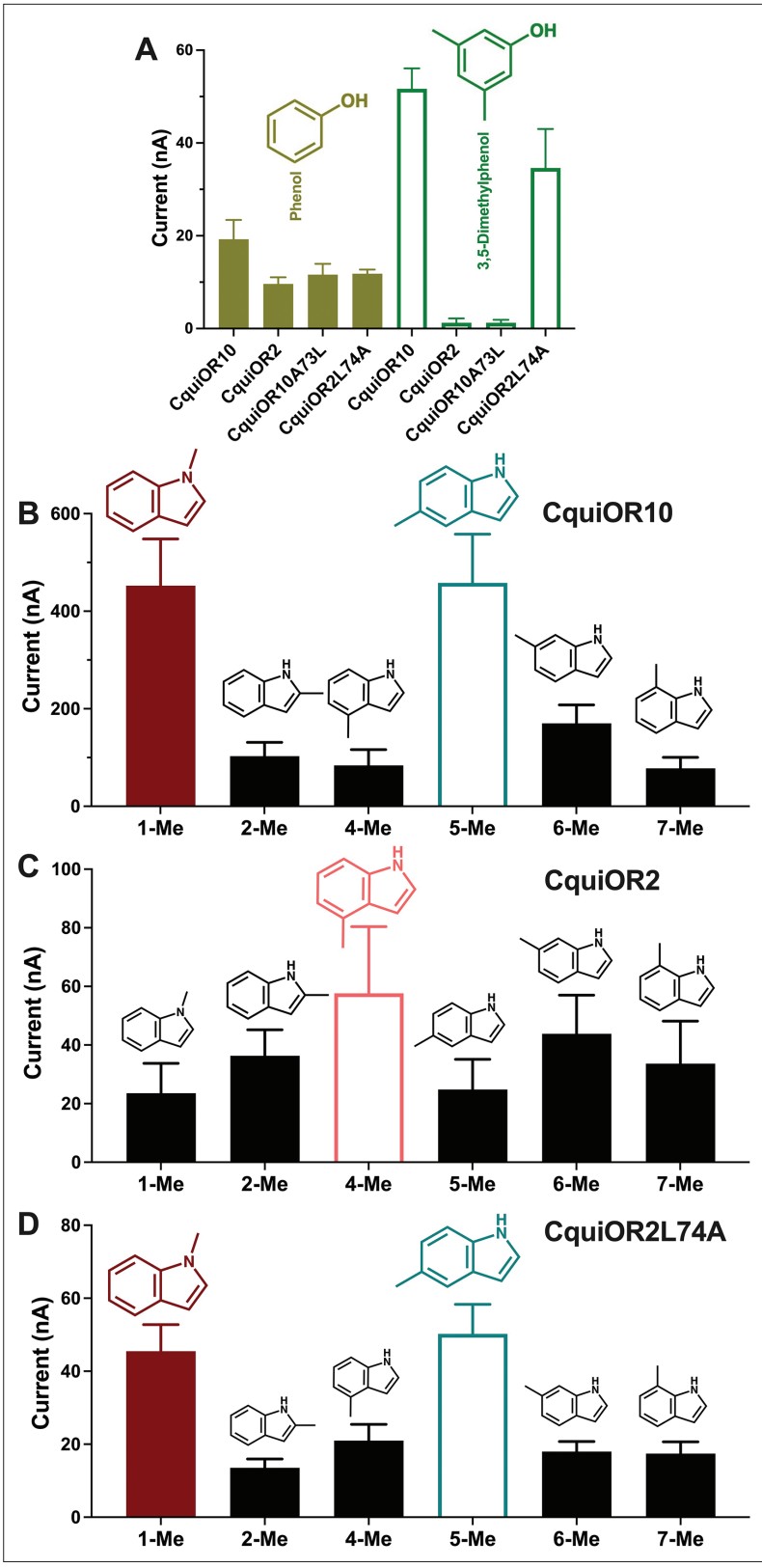

**Figure 5.** Quantification of wildtype and chimeric receptors to phenol and 2,3-dimethylphenol, and methylindoles. (**A**) Each receptor was co-expressed with CquiOrco in *Xenopus* oocytes and stimulated with the phenolic compounds at 1 mM. n = 3–5. (**B**) CquiOR10/CquiOrco-, (**C**) CquiOR2/CquiOrco-, and (**D**)-CquiOR2L74A-expressing oocytes were stimulated with 100 µM of the specified methylindoles. n = 9–11. Bars represent SEM.

*Figure 5 continued on next page*

*Figure 5 continued*

The online version of this article includes the following source data for figure 5:

**Source data 1.** Quantification of wildtype and chimeric receptors to phenol and 2,3-dimethylphenol, and methylindoles.

hypotheses (*Supplementary file 1*, Tables 4 and 5). Our modeling suggests that both indole and skatole can readily reorient themselves in a similar pore depth near residue 73, regardless of mutant or wildtype receptor. Comparing the lowest interface-scoring model from each receptor–ligand complex, indole and skatole are positioned in the membrane-embedded pore, flanked by transmembrane helices S2, S4, S5, and S6, and show positional overlap in both and CquiOR10-A73L CquiOR10 (*Figure 7*, *Figure 7—figure supplements 1–5*).

In most models, skatole and indole form contacts with CquiOR10 and CquiOr10A73L in a similar plane about a center of rotation. These observations are supported by skatole and indole not containing rotatable bonds, thus relying on rigid translational movements and rotation to form favorable contacts with the rotatable and repackable receptor residues. Additionally, our models position indole and skatole within a series of nonpolar, polar-uncharged, and aromatic amino acids. Protein–ligand interaction profiler (PLIP) analysis (*Adasme et al., 2021*) suggests that the bulk of favorable interactions are nonpolar, occasional hydrogen bonding with the odorant NH group, and occasional parallel pi stacking, with the ligand-binding pocket formed by TMs 2, 4, 5, and 6 (*Figure 7—figure supplement 6*). Akin to eugenol forming hydrophobic contacts with MhraOR5/Ile-213 from TM4 (*Figure 6—figure supplement 3*), skatole and indole formed hydrophobic contacts with CquiOR10/Asn-72 from TM2 (*Supplementary file 1*, Table 8), which are matched pairs form Needleman–Wunsch pairwise alignment (*Figure 6—figure supplement 1*). We find of most importance skatole and indole not forming contacts with Ala-73 in CquiOR10 models (*Supplementary file 1*, Tables 6–14). By contrast, in the CquiOR17A73L models, skatole formed hydrophobic contacts with Leu-73 in 5 of the 10 representative models, while indole formed contacts with Leu-73 in 2 representative models (*Supplementary file 1*, Table 8). This suggests that Ala-73 may indirectly affect specificity by modulating the volume of the binding pocket (see Appendix 3).

Structurally aligning CquiOR10 and CquiOR10A73L receptors by TM7b (*Supplementary file 1*, Table 15), demonstrates approximately a 1 Å α-carbon outward shift of A73L (*Supplementary file 1*, Table 16), suggesting a tightly constrained space in CquiOR10 and an expanded space in CquiOR10A73L relative to the protein backbone (*Figure 8*).

This difference in binding pocket volume is not ligand-induced, but rather independent of skatole or indole binding. Consistent with our structural hypothesis of space constraints, mutation of Ala-73 in CquiOR10 to Ile or Val negatively affected receptor function, whereas CquiOR10A73G retained specificity and showed higher sensitivity. Specifically, CquiOR10A73I/CquiOrco- and CquiOR10A73V/CquiOrco-expressing oocytes did not respond to skatole or indole (*Figure 8—figure supplement 1A and B*). We concluded that mutations with these bulkier residues caused loss of binding to indole or skatole, given that these receptors were functional, as indicated by the potent responses elicited by the Orco ligand candidate OLC12, 2-{[4-Ethyl-5-(4-pyridinyl)–4 H-1,2,4-triazol-3-yl]sulfanyl}-*N*-(4-isopropylphenyl)acetamide (*Chen and Luetje, 2012*), also known as VUAA-3 (*Taylor et al., 2012*). As previously demonstrated, OR-Orco complexes are more sensitive to activation by Orco agonists than are the Orco homomers (*Chen and Luetje, 2012*; *Chen and Luetje, 2013*; *Choo et al., 2018*; *Hughes et al., 2017*; *Chen and Luetje, 2014*). It is, therefore, conceivable that the complexes (*Figure 8—figure supplement 1A, B*) were expressed but the binding sites were defective.

On the other hand, CquiOR10A73G/CquiOrco-expressing oocytes showed the same skatole specificity as the wildtype receptor (*Figure 8—figure supplement 1C*). Interestingly, skatole elicited stronger currents recorded from CquiOR10A73G than the wildtype receptor CquiOR10 (*Figure 8—figure supplement 2*), further implying a higher affinity for skatole when there is a reduced constraint, or increased volume, of the binding pocket.

Since CquiOR2 is homologous to CquiOR10, we also propose that our space constraint structural hypothesis can be applied to CquiOR2. CquiOR2 has 49.5% sequence identity and 71.7% sequence similarity to CquiOR10. CquiOR2 also has physiochemically matched pairs to CquiOR10 residues speculated to form contacts with skatole/indole in our study (*Supplementary file 1*, Table 17). While we did not perform CquiOR2 docking, we indirectly examined space constraints in CquiOR2 by

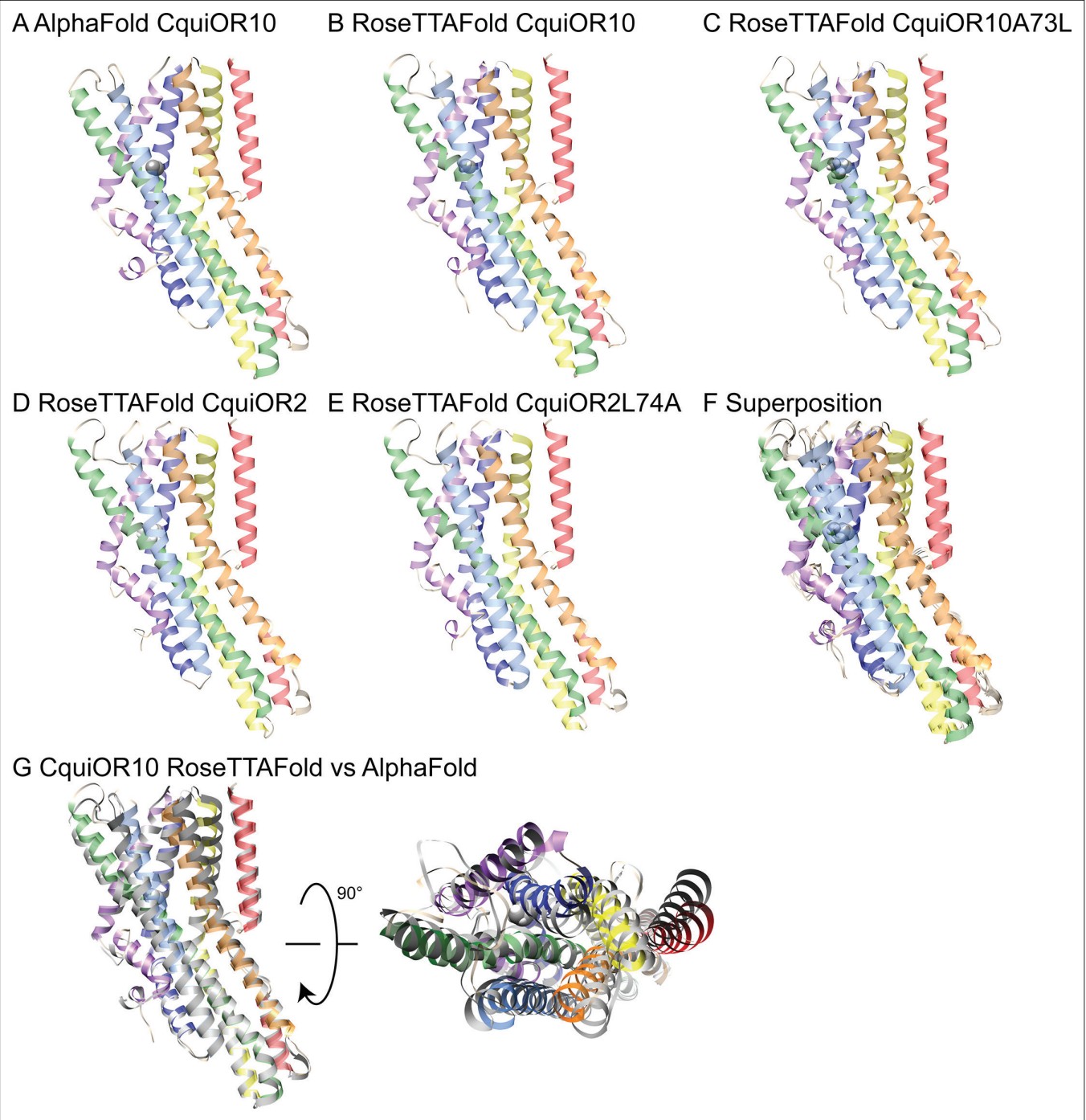

**Figure 6.** AlphaFold and RoseTTAFold models. Structural models of CquiOR10 (**A, B**), CquiOR10A73L (**C**), CquiOR2 (**D**), and CquiOR2L74A (**E**) with AlphaFold (**A**) and RoseTTAFold (**B–E**) structure prediction methods. Superposition of all RoseTTAFold models (**F**) resulted in transmembrane helix root mean square deviation (RMSD) of 0.8 Å when aligned with RoseTTAFold CquiOR10. (**G**) The transmembrane helix RMSD of CquiOR10 RoseTTAFold (rainbow) vs. AlphaFold (gray) was 1.7 Å. Loops were not included in RMSD calculation due to inherent flexibility during structure prediction.

The online version of this article includes the following figure supplement(s) for figure 6:

**Figure supplement 1.** Pairwise alignment of CquiOR10 and MhraOR5.

**Figure supplement 2.** Overlay of MhraOR5 structure and CquiOR10 models.

**Figure supplement 3.** Representative model of docked eugenol with MhraOR5.

**Figure supplement 4.** Additional clusters of eugenol docked to MhraOR5.

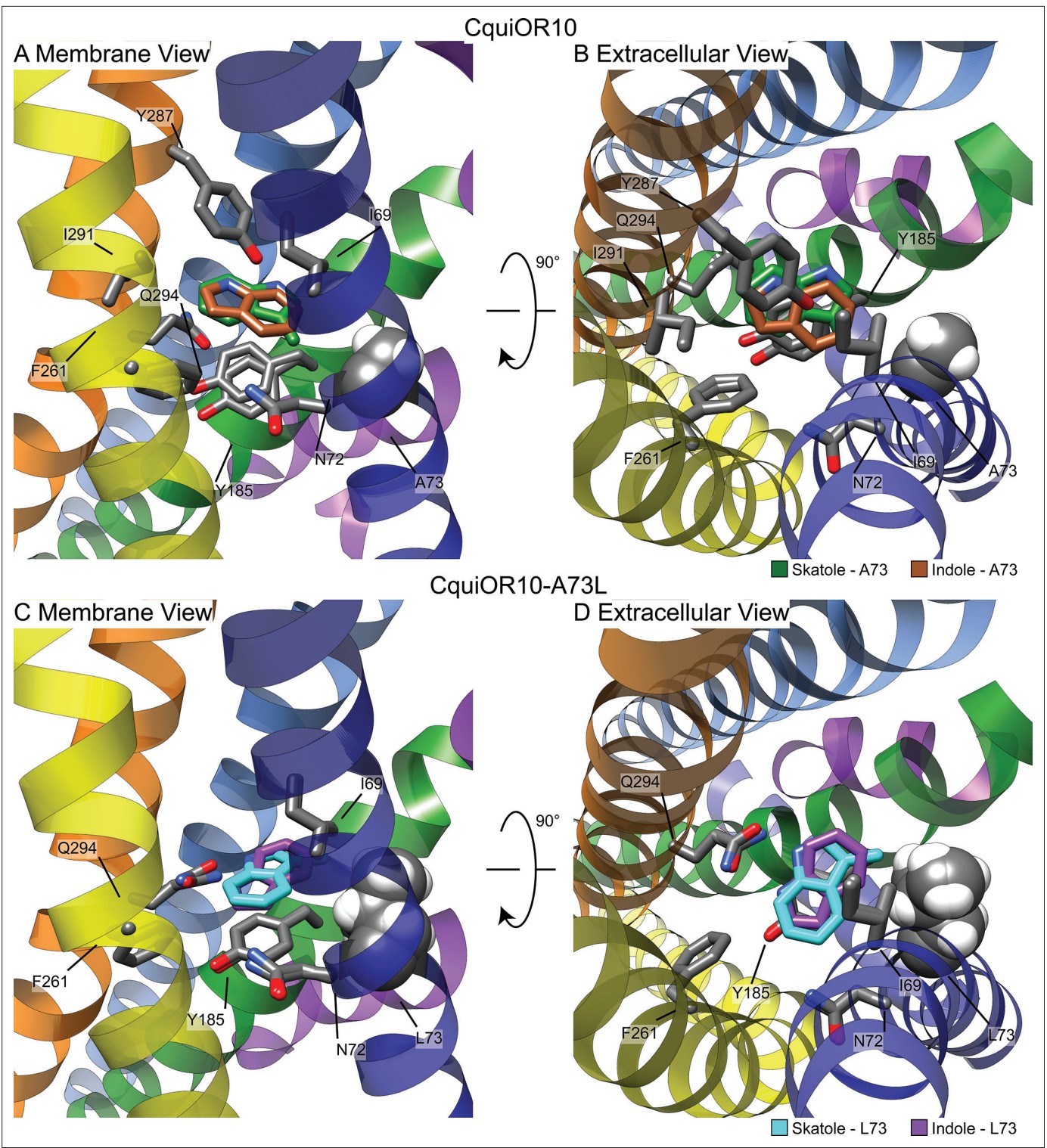

**Figure 7.** Representative models of docked skatole and indole in complex with CquiOR10 and CquiOR10A73L using RosettaLigand. Each model shown is the lowest interface-energy model from the 10 largest clusters of each docking study. CquiOR10 – skatole (forest green), CquiOR10 – indole (brown), CquiOR10A73L – skatole (light blue), and CquiOR10A73L – indole (purple). Atoms that are not indole/skatole carbon atoms are color-coded by atom type: carbon (gray), nitrogen (dark blue), and oxygen (red). Ala-73 and Leu-73 indicated with space-filling representation. (**A, B**) and (**C, D**) Mebrane and extracellular views for CquiOR10 and CquiOR10A73L, respectively.

The online version of this article includes the following figure supplement(s) for figure 7:

**Figure supplement 1.** Clusters from RosettaLigand docking of skatole or indole to CquiOR10 or CquiOR10A73L.

*Figure 7 continued on next page*

*Figure 7 continued*

**Figure supplement 2.** Zoom-out view of CquiOR10 or CquiOR10A73L complexed to skatole or indole.

**Figure supplement 3.** Representative models of docked skatole or indole to CquiOR10 or CquiOR10A73L with PLIC analysis.

**Figure supplement 4.** Superimposition of all OR-skatole and OR-indole clusters.

**Figure supplement 5.** Example of sampling from ligand docking.

**Figure supplement 6.** Representative RosettaLigand docking of skatole and indole to (**A**) CquiOR10 and (**B**) CquiOR10A73L with PLIP analysis.

**Figure supplement 7.** Superposition of all clusters of skatole and indole docked to (**A**) CquiOR10 and (**B**) CquiOR10-A73L.

**Figure supplement 8.** Example of sampling from ligand docking.

mutating Leu-74 to Ile, Val, or Gly. We then tested oocytes co-expressing CquiOrco with CquiOR2L74I, CquiOR2L74V, or CquiOR2L74G (*Figure 8—figure supplement 1D–F*). Consistent with the relaxation of the space constraints, CquiOR2L74I retained the specificity of the wildtype receptor to indole, whereas CquiOR2L74G showed a reverse specificity profile (*Figure 8—figure supplement 1F*), that is, more robust response to skatole than indole. Interestingly, CquiOR2L74V/CquiOrco-expressing oocytes generated nearly equal, albeit small, currents when stimulated with the two oviposition attractants (skatole, 8.7 ± 2.7 nA; indole, 9.0 ± 2.5 nA at 100 μM; n = 3, p>0.9999, Wilcoxon matched-pairs signed-rank test).

Next, we tested the space constraints hypothesis with a bulkier ligand, 3-ethylindole. A CquiOR10 mutant with a less space-filling residue, CquiOR10A73G, elicited responses to 3-ethylindole (526 ± 110 nA) higher than the responses to indole (205 ± 81 nA, at 100 μM), although less robust than the skatole responses (1939 ± 142 nA; all at 100 μM) (*Figure 9—figure supplement 1*). By contrast, receptors with a bulkier residue (CquiOR2WT, CquiO10A73L, and CquiOR2L74I) did not respond to 3-ethylindole (*Figure 9*). However, CquiOR2 mutant with less space-filling residues, CquiOR2L74A and CquiOR2L74G, responded to 3-ethylindole in a dose-dependent manner (*Figure 9*). Additionally, CquiOR10A73G elicited dose-dependent strong responses to 3-ethylindole than CquiOR10 (*Figure 9*).

In summary, the findings that a bulkier, non-natural ligand, elicited more robust responses when CquiOR10 and CquiOR2 residues at 73 and 74, respectively, were mutated into Gly are consistent with the space constraints hypothesis.

## Discussion

Our mutation studies demonstrated that a single amino acid residue substitution in two narrowly tuned ORs from the southern house mosquito can revert their specificity to the oviposition attractants skatole and indole. Amino acid residues leading to one-way alterations of insect ORs have been previously reported (*Auer et al., 2020*; *Cao et al., 2021*; *Hughes et al., 2014*; *Leary et al., 2012*; *Pellegrino et al., 2011*; *Yang et al., 2017*; *Yuvaraj et al., 2021*). Here, we demonstrated that switching a single amino acid residue in two mosquito ORs reverses the specificity of these receptors to two physiologically and ecologically significant odorants. Understanding how mosquito odorant receptors detect oviposition attractants may lead to the development of potent lures. Trapping female that already had a blood meal is an invaluable tool for surveillance given that these gravid females are likely to carry virus circulating in an area.

It has been proven challenging to obtain cryo-EM structure of insect ORs. Hitherto, the only experimentally solved structures of insect receptors are AbakOrco, a co-receptor from the parasitic fig wasp, *A. bakeri* (*Butterwick et al., 2018*), and MhraOR5, an OR from the jumping bristle, *M. hrabei* (*DelMarmol et al., 2021*). While structures of ORs from more evolved winged insects (Pterygota) are yet to be experimentally determined, the most accurate structural modeling methods, such as Rosetta, RoseTTAFold, and AlphaFold, allow us to get a better understanding on how these receptors interact with ligands. Here, we obtained unambiguous experimental evidence that Ala-73 plays a crucial role in CquiOR10 specificity to skatole. We then resorted to modeling to put forward structural hypotheses that can be tested using available experimental approaches and validated by high-resolution structures in the future. Using RoseTTAFold and AlphaFold, we generated models of CquiOR10 followed by RosettaLigand docking of skatole and indole to generate structural hypotheses (*Figure 6*, *Figure 7*, *Figure 7—figure supplements 1–8*). RoseTTAFold (*Baek and Baker, 2022*)

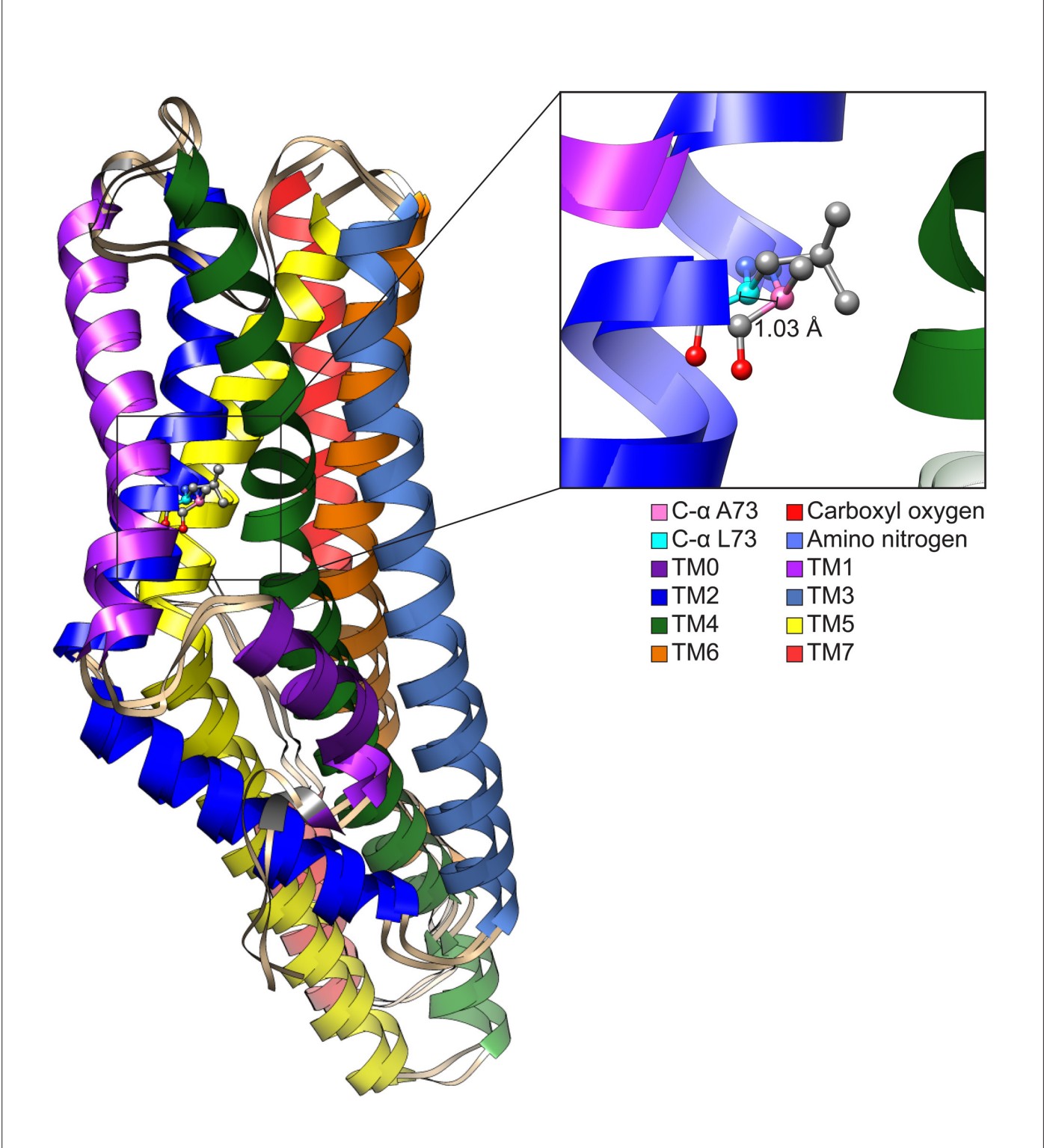

**Figure 8.** Comparison of CquiOR10 and CquiOR10A73L models. An approximate 1 Å α-carbon outward shift of Leu-73 (forest green) in CquiOR10 model relative to Ala-73 (light blue) in CquiOR10A73L model. Models were superimposed using the TM7b region. Residue 73 amino nitrogen is colored in dark blue, and carboxyl oxygen is colored in red in each model.

The online version of this article includes the following figure supplement(s) for figure 8:

**Figure supplement 1.** Effect of single-point mutations around A73.

**Figure supplement 2.** Quantification of single-point mutations around A73.

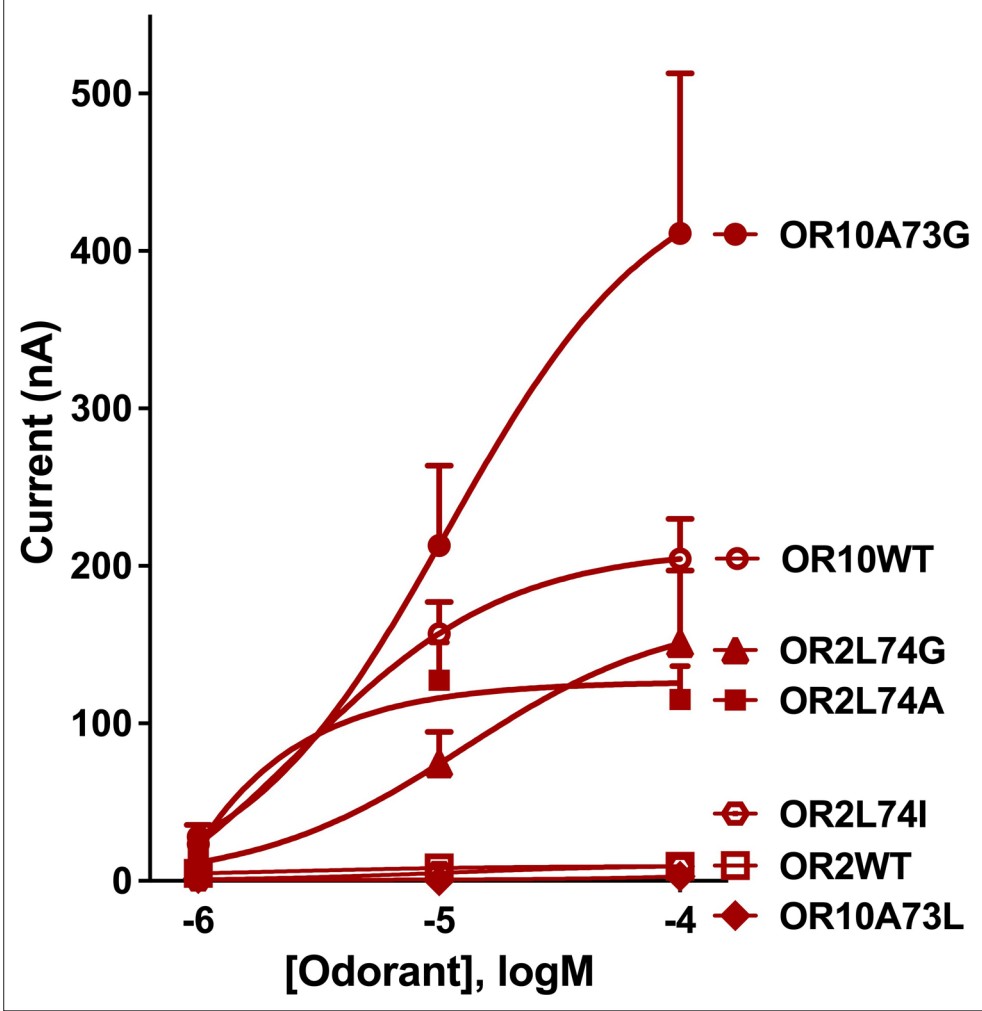

**Figure 9.** Concentration-dependent responses elicited by 3-ethylindole in oocytes co-expressing CquiOrco with CquiOR10, CquiOR2, or single-point mutants. Bars represent SEM (n = 4–10).

The online version of this article includes the following source data and figure supplement(s) for figure 9:

**Source data 1.** Concentration-dependent responses elicited by 3-ethylindole in oocytes co-expressing CquiOrco with CquiOR10, CquiOR2, or single-point mutants.

**Figure supplement 1.** Representative trace of the responses of CquiOR10A73G/CquiOrco-expressing oocyte to indole, skatole, and 3-ethylindole (brown).

and AlphaFold (*Jumper et al., 2021*) are highly accurate and state-of-the-art tools for protein structure prediction. On the other hand, RosettaLigand (*Davis and Baker, 2009*, *DeLuca et al., 2015*) is a competitive protein docking method (*Smith and Meiler, 2020*) that has been used previously to predict the binding position of ligands within protein pores (*Craig et al., 2020*; *Nguyen et al., 2017*; *Nguyen et al., 2019*; *Pressly et al., 2022*). Of note, a structure of an OR-Orco heterocomplex has yet to be elucidated. It has been postulated that they adopt the same overall architecture as the Orco homomeric channel (*Butterwick et al., 2018*), with one or more Orco subunits being replaced by an OR. Additionally, the structure of the 'stand alone' MhraOR5 is remarkably similar in the fold of the helical subunits and the quaternary structure formed by the four subunits within the membrane plane (*DelMarmol et al., 2021*). Therefore, it is reasonable to assume that ligand–receptor interactions analyzed with homomers reflect the same interactions with heteromers.

RoseTTAFold modeling and RosettaLigand docking studies suggest that the specificity determinant amino acid residue (Ala-73) did not form direct contacts with ligands but rather provided a finely tuned, sensitive volumetric space to accommodate odorants. Interpreting CquiOR10-A73L as

an expanded space is counterintuitive compared with our oocyte recordings; the wild-type CquiOR10 receptor responded better to the bulkier skatole (*Figure 1A*), while CquiOR10-A73L responded better to the smaller indole (*Figure 2I*).

Considering both experimental and modeling data, the reason for differing receptor responses at residue 73 could be due to the repacking of this position to accommodate more carbon side-chain atoms that are reducing the binding pocket volume, or shift key, nearby residues required for receptor response. The fact that indole is a rigid molecule, and that the methyl group of skatole is the only rotamer, could also suggest that the binding pocket around position 73 is tightly regulated. The addition of a methyl group could prevent skatole from occupying the appropriate configuration for receptor response with Leu-73 constrained volume (Appendix 4). Combined, our results suggest that the residue at 73 provides a finely tuned volumetric space to accommodate specific oviposition attractants. Future structures can test the space constraints hypothesis proposed here by validating the specific residues interacting with odorants, quantifying the binding pocket volume, and providing structural insight into how position 73 modulates receptor response to specific odorants.

Taken together, our findings shed light on a possible path to design more potent oviposition attracts to trap mosquitoes, which may pave the way to novel strategies for vector-borne virus surveillance and, possibly, mosquito control.

# Materials and methods

## Key resources table

| Reagent type (species) or resource | Designation | Source or reference | Identifiers | Additional information |
|---|---|---|---|---|
| Recombinant DNA reagent | Stellar competent cell | Takara Bio, USA (San Jose, CA) | Cat# 636766 | https://bit.ly/3Dowpe2 |
| Recombinant DNA reagent | pGEMHE (plasmid) | *Liman et al., 1992* | https://doi.org/10.1016/0896-6273(92)90,239a | |
| Recombinant DNA reagent | *Xenopus* oocytes | EcoCyte Bioscience (Austin, TX) | https://bit.ly/3Ud8OTo | |
| Recombinant DNA reagent | XmaI | New England Biolabs (Ipswich, MA) | Cat# R0180S | https://www.neb.com/products/r0180-xmai |
| Recombinant DNA reagent | XbaI | New England Biolabs (Ipswich, MA) | Cat# R0145S | https://www.neb.com/products/r0145-xbai |
| Recombinant DNA reagent | Gentamycin sulfate | Abcam (Cambridge, UK) | Cat# ab146573 | https://www.abcam.com/ab146573.html |
| Chemical compound, drug | NaCl | Fisher Scientific (Waltham, MA) | Cat# S271-3 | |
| Chemical compound, drug | KCl | Fisher Scientific (Waltham, MA) | Cat# P217-500 | |
| Chemical compound, drug | $NaHCO_3$ | Sigma-Aldrich (Milwaukee, WI) | Cat# S6014-500G | |
| Chemical compound, drug | $MgSO_4$ | Sigma-Aldrich (Milwaukee, WI) | Cat# M-7634 | |
| Chemical compound | $Ca(NO_3)_2$ | Sigma-Aldrich (Milwaukee, WI) | Cat# 237124-500G | |
| Chemical compound, drug | $CaCl_2$ | Fisher Scientific (Waltham, MA) | Cat# S71924 | |
| Chemical compound, drug | HEPES | Sigma-Aldrich (Milwaukee, WI) | Cat# H4034-500G | |
| Chemical compound, drug | OLC12 | Vanderbilt Institute of Chemical Biology | Chemical Synthesis Core, VUAA 3 | https://medschool.vanderbilt.edu/syncore/ |
| Chemical compound, drug | Skatole | Sigma-Aldrich (Milwaukee, WI) | CAS# 83-34-1, Cat# W301912 | 98% |
| Chemical compound, drug | Indole | ACROS Organics (Geel, Belgium) | CAS# 120-72-9, Cat# 122150100 | 98% |
| Chemical compound, drug | 3-Ethylindole | AmBeed (Arlington hts, IL) | CAS# 1484-19-1, Cat# AMBH96F1079C | 97% |
| Chemical compound, drug | Phenol | Sigma-Aldrich (Milwaukee, WI) | CAS# 108-95-2 | 99.5% |
| Chemical compound, drug | 3,5-Dimethylphenol | Sigma-Aldrich (Milwaukee, WI) | CAS# 108-68-9 | 99% |
| Chemical compound, drug | 1-Methylindole | Sigma-Aldrich (Milwaukee, WI) | CAS# 603-76-92 | 97% |

*Continued on next page*

*Continued*

| Reagent type (species) or resource | Designation | Source or reference | Identifiers | Additional information |
|---|---|---|---|---|
| Chemical compound, drug | 2-Methylindole | Sigma-Aldrich (Milwaukee, WI) | CAS# 95-20-5 | 98% |
| Chemical compound, drug | 4-Methylindole | ACROS Organics (Geel, Belgium) | CAS# 16096-32-5 | 99% |
| Chemical compound, drug | 5-Methylindole | Sigma-Aldrich (Milwaukee, WI) | CAS# 614-96-0 | 99% |
| Chemical compound, drug | 6-Methylindole | Sigma-Aldrich (Milwaukee, WI) | CAS# 3420-02-8 | 97% |
| Chemical compound, drug | 7-Methylindole | Sigma-Aldrich (Milwaukee, WI) | CAS# 933-67-5 | 97% |
| Software, algorithm | UCSF Chimera | *Pettersen et al., 2004* | https://doi.org/10.1002/jcc.20084; UCSF | https://bit.ly/3S7OdOF; ver. 1.15 |
| Software, algorithm | Rosetta | *Leman et al., 2020* | https://doi.org/10.1038/s41592-020-0848-2 | https://www.rosettacommons.org/software/license-and-download; ver 2021.07.61567 |
| Software, algorithm | Avogadro | *Hanwell et al., 2012* | https://doi.org/10.1186/1758-2946-4-17 | https://avogadro.cc/; ver 1.2.0 (Git revision: c1fcc5b) |
| Software, algorithm | AmberTools | *Case et al., 2021* | https://ambermd.org/index.php | https://ambermd.org/doc12/Amber21.pdf |
| Software, algorithm | OpeneEye Omega | *Hawkins et al., 2010* | https://doi.org/10/1021/ci100031x | https://www.eyesopen.com/omega |
| Software, algorithm | HDBSCAN | *McInnes et al., 2017* | https://doi.org/10.21105/joss.00205 | https://github.com/scikit-learn-contrib/hdbscan; *McInnes and Healy, 2017* |
| Software, algorithm | BioMol2Clust | https://biokinet.belozersky.msu.ru/Biomol2Clust | *Timonina et al., 2021* | ver 1.3 |
| Software, algorithm | Protein Ligand Interaction Profiler | *Salentin et al., 2015* | https://doi.org/10.1093/nar/gkv315 | https://plip-tool.biotec.tu-dresden.de/plip-web/plip/index; software repository: https://github.com/pharmai/plip; ver 2.2.1, *Salentin et al., 2015* |
| Software, algorithm | EMBOSS Needle | *Madeira et al., 2022* | https://doi.org/10.1093/nar/gkac240 | https://www.ebi.ac.uk/Tools/psa/emboss_needle/ |

## Construction of chimeric receptors

We used a previously obtained pGEMHE-CquiOR10 (*Hughes et al., 2010*) plasmid to amplify the sequences of the desired chimeric receptors. To obtain the full length of chimeric *OR* genes, we used specific primers, with overlapping sequences at each end to target sequences (adaptor underlined) for cloning into pGEMHE vector, which was linearized at restriction sites *XmaI* (5′-CCCGGG-3') and *XbaI* (5′-TCTAGA-3'):CqOR10_Fw 5′-GATCAATTCCCCGGGACCATGACCGCGGCACCCATTTT-3' and CqOR10_Rv 5′-CAAGCTTGCTCTAGATCAATTATAAACGCGTCTCAGCAGGGT-3'. To generate a chimeric OR by PCR amplification, we designed specific primers for the desired CquiOR2 TM domain. Simultaneously, we prepared the recipient fragments of CquiOR10 to receive the CquiOR2 TM domains. The fragments were then assembled using In-Fusion HD cloning kit (Clontech) to obtain the desired chimeric OR. Each CquiOR2 TM domain was divided into two parts, each part was synthesized with a specific primer, except for CquiOR2 TM7 domain, which used only one primer with the full domain. These primers contained an overlap with CquiOR10 sequence followed by one part of the CquiOR2 TM sequence and an overlap with the other part of the CquiOR2 TM domain sequence, which was synthesized in another PCR. The underlined sequences represent CquiOR2 TM domains and, unless otherwise specified, sequences in italic are overlaps for CquiOR2 TM domains. CqOR10^M1_ Fw: 5′-GGTGACCGTGCTGAACGTGTTCCAGTTTATGAACCTGTTTCGAGCCTGGGGCAACATC-3'; CqOR10^M1_Rv: 5′-TCAGCACGGTCACCGGAATGCAGCCGAGGAGGTAACTGAGGACGTTGCCTACTGTGATCTCAAGG-3'; CqOR10^M2_Fw: 5′-CCGTGCTGTACTTCAACCTTGTGTTGAGAACCACGTTTATACTGTGCAATCGTCAGGATTATGAGG-3'; CqOR10^M2_Rv: 5′-*AAGTACAGCACGGTGAAATATCCGTCGATGATGATTTT*GTCGATGTTGCCCCAGGCT-3'; CqOR10^M3_Fw: 5′-*GTTCATCAGTGCGTG*CTTCGTGACGTATCCGCTTTTTTCACCGACACGTAGCCTCCCGTACG-3'; CqOR10^M3_Rv: 5′-*ACGCACTGATGAAC*GCTCCCAGCCAGAGGTTCGATTTGGACAGCAGTCGGGCACGTTTGGTGA-3'; CqOR10^M4_Fw: 5′-*CACGTTTCCGGCGTG*CTGCATGTACATTCCGTTTACCAGCTTCTTCGCCACGACTACTTTG-3'; CqOR10^M4_Rv: 5′-*ACGCCGGAAACGTG*AGGTACACTTGCAGAAAAAACACAACC

TGGTACAGGGGCGTC-3'; CqOR10$^{M5}$_Fw: 5'- *GCTATGCGCCTTGC*TGTTTCTACTTAGCACCAGCAAT AATCCCGCGCAAATTATCATCGTGG-3'; CqOR10$^{M5}$_Rv: 5'- *GCAAGGCGCATAGC*ATCATCCCAAAC GATAGCAACTCAATCAGACAGATGTAGGTCACCAGCG-3'; CqOR10$^{M6}$_Fw: 5'- *TCTTTATGATTCT* GTCCCAGATGTACGCCCTGTACTGGCACGCCAACGAGCTGCG-3'; CqOR10$^{M6}$_Rv: 5'- *AGAATCAT AAAGA*TGTACGATCCGATCATCACCATCTGCGCGGGATTTTCGATAATGTTCAGC-3'; CqOR10$^{M7}$_Rv: 5'- *CAAGCTTGCTCTAGA*TCAATTATAAACGCGTCTCAGCAACGTAAAGTACGAATACGAGGCATTG ATCAACTTTTGAAACATTTCCAAGGTCATCGGATAGACGTTGCCTACTGTGATCTCAAGG-3' (here the italic represents the pGEMHE adaptor). The two fragments were amplified using a combination of CqOR10_Fw with CqOR10$^{Mx}$_Rv primers and CqOR10_Rv with CqOR10$^{Mx}$_Fw primers. Then, we cloned the two fragments amplified by PCR into pGEMHE vector using the In Fusion system as described below. Chimeric plasmid with a single TM swapped was used as a template to generate chimeric OR with two TM swapped and subsequently chimeric ORs with multiple CquiOR2 TM domains.

## Chimeric OR cloning and subcloning into pGEMHE

The fragments amplification was performed using Platinum Taq DNA Polymerase High Fidelity (Thermo Fisher Scientific) with the following conditions: 95°C for 5 min, followed by 30 cycles of 95°C for 20 s, 57°C 30 s for annealing and 68°C for 1.5 min, and extension at 68°C for 5 min. PCR products were purified by QIAquick gel extraction kit (QIAGEN). The target pGEMHE plasmid was cut by *XmaI* and *XbaI* in separate reactions. Ligation was done with In Fusion (Takara Bio USA) system and the transformation was performed using Stellar competent cells (Takara Bio USA) in heat-shock. After selecting the cells, plasmids were extracted with QIAprep Spin Miniprep kit (QIAGEN). The cloned gene was verified by DNA sequencing (Berkeley Sequencing Facility).

## Site-directed point mutagenesis and fragment replacement

Phusion Site-Directed Mutagenesis Kit (Thermo Scientific, West Palm Beach, FL) was used to generate point mutations and TM fragment replacements. Mutations were created with mismatched 5'-phosphorylated mutagenic primers and PCR amplification. The chimeric sequences (CquiOR10$^{M2,5,6,7}$) in pGEMHE vector were used as templates for rescues, whereas the wildtype sequences in pGEMHE vector served as templates for point-mutations. Rescue primers: CqOR10M2_M2567OUTERup: 5'-G ATGATGACCTCGTCGATGTTGCCCCAGGC-3'; CqOr10M2_M2567OUTERdn: 5'-AACGCATATTTCA CCGTGCTGTACTTCAACC-3'; CqOr10M2_M2567INNERup: 5'-cgcagcaccgcgttgaagtacagcacggt g-3'; CqOr10M2_M2567INNERdn: 5'-aacaattttcatactgtgcaatcgtcagga-3'; CqOR10M2_M2567MID-INNERup: 5'-catcgacggCtactttgcgatgattttcttcaacgcg-3'; CqOR10M2_M2567MID-INNERdn: 5'-a tgattttgtcgatgttgccccaggctc-3'; CqOr10M2_M2567_L73Aup: 5'-CGTTGAAGTACAGCACGGTGA AATAT-3'; CqOr10M2_M2567_L73Adn: 5'-CTGTGTTGAGAACCACGTTTATACTGT-3'; CqOr10M2_ M2567_T78Iup: 5'-ATGGTTCTCAACACAAGGTTGAAGTA-3'; CqOr10M2_M2567_T78Idn: 5'-CTTTA TACTGTGCAATCGTCAGGATTATG-3'; point mutation primers: CqOR10A73up: 5'-Gttgaagaaaatc atcgcaaagta-3'; CqOR10A73Ldn: 5'-Ctggtgctgcgaacaattttc-3'; CqOR10A73Adn: 5'-Gcggtgctgcgaa caattttc-3'; CqOR10A73Gdn: 5'-Ggggtgctgcgaacaattttc-3'; CqOR10A73Vdn: 5'-Gtggtgctgcgaacaat tttc-3'; CqOR10A73Idn: 5'-ATCgtgctgcgaacaattttc-3'; CqOR2L74up: 5'-gttgaagtacagcacggtgaaata-3'; CqOr2L74Adn: 5'-Gctgtgttgagaaccacgtttatac-3'; CqOr2L74Idn: 5'-atcGTGTTGAGAaccacgtttatac-3'; CqOr2L74Gdn: 5'-gggGTGTTGAGAaccacgtttatac-3'; CqOr2L74Vdn: 5'-gtgGTGTTGAGAaccac gtttatac-3'. The amplified linear PCR products containing the desired modification were ligated and transformed into Stellar Competent Cells (Takara Bio USA). All sequences were confirmed by DNA sequencing (UC Berkeley DNA Sequencing Facility).

## In vitro transcription, oocyte microinjection, and two-electrode voltage-clamp assay (TEVC)

Capped OR cRNA was prepared using mMESSAGE mMACHINE T7 Kit (Ambion) as previously described (*Xu et al., 2019*). Purified OR cRNA was resuspended in nuclease-free water at 200 ng/ µL and microinjected into *Xenopus laevis* oocytes on stage V or VI (EcoCyte Bioscience) along with the same amount of CquiOrco. Injected oocytes were incubated at 18°C for 3–7 days in a modified Barth's solution (88 mM NaCl, 1 mM KCl, 2.4 mM NaHCO$_3$, 0.82 mM MgSO$_4$, 0.33 mM Ca(NO$_3$)$_2$, 0.41 mM CaCl$_2$, and 10 mM HEPES at pH 7.4) supplemented with 10 µg/mL gentamycin, 10 µg/

mL streptomycin, and 1.8 mM sodium pyruvate. Two-electrode voltage-clamp technique (TEVC) was used to measure odorant-induced currents at a holding potential of −80 mV. Signals were amplified with an OC-725C amplifier (Warner Instruments), low-pass filtered at 50 Hz, and digitized at 1 kHz. Data acquisition and analysis were performed with Digidata 1440A and software pCLAMP 10 (Molecular Devices). Skatole (98%, CAS# 83-34-1), indole (98%, CAS# 120-72-9), and 3-ethylindole (97%, CAS# 1484-19-1) were provided by Sigma-Aldrich (Milwaukee, WI), ACROS Organics (Geel, Belgium), and AmBeed (Arlington Hts, IL), respectively. Phenol (99.5%, CAS# 108-95-2), 3,5-dimethylphenol (99%, CAS# 108-68-9), 1-methylindole (97%, CAS# 603-76-9), 2-methylindole (98%, CAS# 95-20-5), 5-methylindole (99%, CAS# 614-96-0), 6-methylindole (97%, CAS# 3420-02-8), and 7-methylindole (97%, CAS# 933-67-5) were acquired from Sigma-Aldrich, and 4-methylindole (99%, CAS# 16096-32-5) was provided by ACROS.

## Rosetta structural modeling and docking

We used RoseTTAFold (*Baek et al., 2021*) and AlphaFold (*Jumper et al., 2021*) to generate predicted structures of CquiOR10, CquiOR2, CquiOR10A73L, and CquiOR2L74A monomers. The OpenEye Omega toolkit (*Hawkins et al., 2010*) was used to generate conformer libraries. Molecular docking was performed using RosettaLigand (*Davis and Baker, 2009*; *DeLuca et al., 2015*). The top models were visually analyzed using UCSF Chimera (*Pettersen et al., 2004*).

To verify RosettaLigand could effectively sample homologous receptors in complex with odorants, we used the structure of eugenol in complex with the insect olfactory receptor OR5 from *M. hrabei* (MhraOR5) as a control test case (*DelMarmol et al., 2021*). After verifying RosettaLigand could recapitulate the MhraOR5–eugenol complex from structural studies, we docked indole and skatole to the RoseTTAFold monomer models of CquiOR10 and CquiOR10A73L receptors with the same method. Briefly, the initial placement of all ligands into respective structures was guided by the position of eugenol in a complex with MhraOR5 as a basis since it is the only homologous structure with an odorant. For an unbiased receptor sampling, an initial transformation was performed on the ligand relative to the receptor site using Rosetta's Transform mover. Within a 7 Å sphere, the ligand underwent a Monte Carlo simulation, whereby the ligand was allowed to translate up to 0.2 Å and rotate up to 20°. This was performed 500 times on the ligand, with the lowest scoring pose used as the starting pose for docking. This form of docking is termed local docking and is commonplace when there is experimental evidence and homologous structures as templates. physiologically relevant binding modes can be identified with local, high-resolution ligand docking (*Kaufmann et al., 2009*). For additional methods, see Appendix 2.

## Acknowledgements

We thank Drs. David K Wilson, Enoch Baldwin (UC Davis), and members of our research groups for enlightening discussions. The Orco ligand candidate OLC12 was provided by the Vanderbilt Institute of Chemical Biology, Chemical Synthesis Core, Vanderbilt University. Research reported in this publication was supported by the National Institute of Allergy and Infectious Diseases (NIAID) of the National Institutes of Health under award number R01AI095514. The content is solely the responsibility of the authors and does not necessarily represent the official views of the NIH.

## Additional information

### Funding

| Funder | Grant reference number | Author |
| --- | --- | --- |
| National Institute of Allergy and Infectious Diseases | R01AI095514 | Walter S Leal |

The funders had no role in study design, data collection and interpretation, or the decision to submit the work for publication.

## Author contributions
Flavia P Franco, Formal analysis, Investigation, Writing – review and editing; Pingxi Xu, Vladimir Yarov-Yarovoy, Conceptualization, Formal analysis, Investigation, Writing – review and editing; Brandon J Harris, Formal analysis, Investigation, Writing – original draft, Writing – review and editing; Walter S Leal, Conceptualization, Formal analysis, Funding acquisition, Investigation, Writing – original draft, Project administration

## Author ORCIDs
Flavia P Franco ⓘ http://orcid.org/0000-0003-3739-8625
Vladimir Yarov-Yarovoy ⓘ http://orcid.org/0000-0002-2325-4834
Walter S Leal ⓘ http://orcid.org/0000-0002-6800-1240

## Decision letter and Author response
Decision letter https://doi.org/10.7554/eLife.82922.sa1
Author response https://doi.org/10.7554/eLife.82922.sa2

# Additional files

## Supplementary files
• MDAR checklist

• Supplementary file 1. Needleman-Wunsch percent identity and similarity of CquiOR10 to CquiOR2 transmembrane segments as defined by OCTOPUS.

• Supplementary file 2. Protocol Capture.

## Data availability
All data generated or analysed during this study are included in the manuscript and supporting files. Source data files have been provided for Figures 1, 3, 5, and 9.

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

## Appendix 1

### Preliminary screening of chimeric receptors

CquiOR10$^{M1,2,3,4,5,6,7}$/CquiOrco-expressing oocytes did not respond to skatole or indole. Likewise, the chimeric receptor with six transmembrane domains replaced, CquiOR10$^{M1,2,3,4,5,6}$ was also silent. Next, we replaced one TM at a time. CquiOR10$^{M1}$ and CquiOR10$^{M2}$ were silent, whereas CquiOR10$^{M3}$ (*Figure 1—figure supplement 1A*), CquiOR10$^{M4}$ (*Figure 1—figure supplement 1B*), CquiOR10$^{M5}$ (*Figure 1—figure supplement 1C*), CquiOR10$^{M6}$ (*Figure 1—figure supplement 1D*), and CquiOR10$^{M7}$ (*Figure 1—figure supplement 1E*) responded to both skatole and indole with specificity similar to that of the wildtype receptor. Although these chimeric receptors differed from CquiOR10 in terms of sensitivity, they showed the same specificity to skatole as CquiOR10 (*Figure 1A*). While CquiOR10$^{M3}$ and CquiOR10$^{M4}$ were less sensitive than CquiOR10, CquiOR10$^{M7}$ showed comparable sensitivity to CquiOR10, and CquiOR10$^{M5}$ and CquiOR10$^{M6}$ were more sensitive to skatole than the wildtype receptor. We remained focused on specificity and did not attempt to quantify the different responses.

We tested a chimeric receptor with the other five TM domains swapped. CquiOR10$^{M3,4,5,6,7}$/CquiOrco-expressing oocytes mirrored the profiles obtained with CquiOR10 in terms of sensitivity and specificity (*Figure 1—figure supplement 1F*). We then kept the wildtype TM7, in addition to TM1 and TM2, in the next chimeric ORs tested. Albeit less sensitive than CquiOR10, CquiOR10$^{M3,4,5,6}$ showed the same specificity of the wildtype receptor (*Figure 1—figure supplement 1G*). Following, we tested chimeric receptors with three TM domains swapped. CquiOR10$^{M4,5,6}$ (*Figure 1—figure supplement 1H*), CquiOR10$^{M3,5,6}$ (*Figure 1—figure supplement 1I*), CquiOR10$^{M3,4,6}$ (*Figure 1—figure supplement 1J*), and CquiOR10$^{M3,4,5}$ (*Figure 1—figure supplement 1K*) did not differ from CquiOR10 in terms of specificity or sensitivity. Next, we focused on chimeric receptors with two TMs swapped. Thus, we tested CquiOR10$^{M3,4}$ (*Figure 1—figure supplement 1L*), CquiOR10$^{M3,5}$ (*Figure 1—figure supplement 1M*), CquiOR10$^{M3,6}$ (*Figure 1—figure supplement 1N*), CquiOR10$^{M3,7}$ (*Figure 1—figure supplement 1O*), CquiOR10$^{M4,5}$ (*Figure 1—figure supplement 1P*), CquiOR10$^{M4,6}$ (*Figure 1—figure supplement 1Q*), CquiOR10$^{M4,7}$ (*Figure 1—figure supplement 1R*), CquiOR10$^{M5,6}$ (*Figure 1—figure supplement 1S*), CquiOR10$^{M5,7}$ (*Figure 1—figure supplement 1T*), and CquiOR10$^{M6,7}$ (*Figure 1—figure supplement 1U*). They all showed skatole-specific profiles, most of them with comparable sensitivity, whereas CquiOR10$^{M4,5}$, CquiOR10$^{M5,7}$, and CquiOR10$^{M6,7}$ were more sensitive and CquiOR10$^{M3,6}$ less sensitive than CquiOR10. Interestingly, none of the chimeric receptors having TM1 as one of the two domains swapped was functional. Specifically, oocytes co-expressing CquiOrco with either CquiOR10$^{M1,2}$, CquiOR10$^{M1,3}$, CquiOR10$^{M1,4}$, CquiOR10$^{M1,5}$, CquiOR10$^{M1,6}$, or CquiOR10$^{M1,7}$ did not respond to skatole or indole (*Supplementary file 1*, Table 2). Likewise, CquiOR10$^{M2,3}$, CquiOR10$^{M2,4}$, CquiOR10$^{M2,5}$, and CquiOR10$^{M2,6}$ did not respond to these oviposition attractants (*Supplementary file 1*, Table 2).

## Appendix 2

### Ligand conformer library generation

The Protein DataBank file (.pdb) of eugenol was extracted from the MhraOR5 structure (PDB ID: 7LID), while indole and skatole were extracted as Structure-Data Files files (.sdf) from PubChem (*Kim et al., 2021*). Each ligand structure was bond-corrected, protonated at pH 7.4, energy minimized using the Merck Molecular Force Field (*Halgren, 1996*), and saved as Tripos Mol2 files (.mol2) with the Avogadro software (*Hanwell et al., 2012*). Next, partial charge, atom, and bond-type assignment were performed with Antechamber AM1BCC correction using Amber Tools (*Case et al., 2021*; *Salomon-Ferrer et al., 2013*); AM1BCC partial-charge correction is common in Rosetta ligand docking protocols (*Park et al., 2021*) and has been shown to be similarly ranked with other Rosetta-based partial-charge methods for RosettaLigand (*Smith and Meiler, 2020*). An in-house script using the OpenEye Omega toolkit and Rosetta was used to generate the conformer library and associated Rosetta-readable ligand parameters file (*Supplementary file 2*, Protocol Capture).

### Protein receptor preparation

The MhraOR5 structure was relaxed with backbone constraints using the RosettaRelax protocol (*Nivón et al., 2013*) prior to RosettaLigand docking. This is commonplace to allow the repacking of protein sidechains and minimization of the structure into the Rosetta score function for comparison between structures. For RoseTTAFold-generated structures, the RosettaRelax protocol was not performed since the Rosetta score function was used to generate the structures.

### RosettaLigand docking

RosettaLigand docking was performed using RosettaScripts protocols described previously (*Davis and Baker, 2009*; *DeLuca et al., 2015*). Briefly, initial placement of all ligands into respective structures was guided by the position of eugenol in complex with MhraOR5 as a basis. For an unbiased sampling of the receptor, an initial transformation was performed on the ligand relative to the receptor site using Rosetta's Transform mover. Within a 7 Å sphere, the ligand underwent a Monte Carlo simulation, whereby the ligand was allowed to translate up to 0.2 Å and rotate up to 20°. This was performed 500 times on the ligand, with the lowest scoring pose used as the starting pose for docking. Six cycles of high-resolution docking were performed with repacking of sidechains every third iteration using Rosetta's HighResDocker mover. Lastly, the protein/ligand complex was minimized using Rosetta's FinalMinimizer mover and interface scores reported using the InterfaceScoreCalculator.

For every test case, 100,000 docking poses were generated. The top 10,000 models were selected by Rosetta interface energy (Interface_delta_X) metric, followed by ligand clustering using Biomol2Clust (*Timonina et al., 2021*), the hdbscan algorithm (*McInnes et al., 2017*), and a minimum cluster threshold of 50 poses. The 10 most frequently sampled clusters were visually analyzed using UCSF Chimera (*Pettersen et al., 2004*) by using the lowest interface energy pose in each cluster as the representative for that cluster. The Protein Ligand Interaction Profiler (*Salentin et al., 2015*) was performed on these representative poses to identify potential hydrophobic interactions, hydrogen bonds, π-stacking, and salt bridges. Hydrogen bond and pi-stacking interactions were filtered by previously reported bond distances (*Bissantz et al., 2010*) to present a conservative estimate of potential contacts. Atom types from PLIP analysis are reported using IDATM nomenclature (*Meng and Lewis, 1991*).

## Appendix 3

The ligand-binding pocket is formed by TMs 2, 4, 5, and 6. The most frequent hydrophobic contacts were formed by Tyr-185 (38/40 models; TM4) and Ile-69 (32/40 models; TM2) (*Supplementary file 1*, Tables 6–8). Phe-261 (TM5) also formed hydrophobic interactions in all four receptor–ligand complex sets of models, but less frequently (22/40 models). Asn-72 (TM2), Tyr-287 (TM6), and Ile-291 (TM6) formed hydrophobic contacts more frequently with skatole structures compared to indole structures (Asn-72: 12/20 vs 9/20, Tyr-287: 14/20 vs 5/20, Ile-291: 8/20 vs 3/20). Notably, Leu-73 (TM2) in CquiOR10A73L formed hydrophobic contacts with indole (2/10) and skatole (5/10), but Ala-73 (TM2) did not form hydrophobic contacts with indole or skatole. Tyr-138 (TM3) formed hydrophobic contacts with skatole in CquiOR10A73L (2/10), but not in CquiOR10. Gln-294 (TM6) formed hydrophobic contacts with skatole and indole in these structures, except CquiOR10A73L-skatole; however, Gln-294 does form a hydrogen bond in 1–2 models for each structure. Of note, the range of PLIP-reported hydrophobic contacts among the 10 representative models for each structure changed from an average of $6.3 \pm 0.5$ contacts (mean ± SEM) in CquiOR10 to $5.2 \pm 0.3$ contacts in CquiOR10A73L (p=0.035, Mann–Whitney test). Considering the number of unique residues forming hydrophobic contacts, the averages were $4.9 \pm 0.4$ and $4.9 \pm 0.2$ contacts, respectively (p=0.97, Mann–Whitney test).

There were at most two models in each structure with hydrogen bonds (*Supplementary file 1*, Tables 9–11). For CquiOR10-skatole, two models formed hydrogen bonds with Gln-294. For CquiOR10A73L-skatole, one model formed hydrogen bonds with Asn-72, and one other with Gln-294 (*Supplementary file 1*, Table 11). For CquiOr10-indole, one model formed a hydrogen bond with the backbone of Ile-69, another model with Tyr-138, another with Tyr-287, and a fourth model with Gln-294. For CquiOR10A73L-indole, only one model formed a hydrogen bond at Gln-294.

Parallel pi-stacking interactions (*Supplementary file 1*, Tables 12–14) were identified for CquiOR10A73L-skatole, CquiOR10-indole, and CquiOR10A73L-indole. For CquiOR10A73L-skatole, two models each formed a parallel pi-stacking interaction with Tyr-185 (*Supplementary file 1*, Table 14). For CquiOR10-indole, a model formed a parallel pi-stacking interaction with Tyr-138 and another model formed with Tyr-152. For CquiOR10A73L-indole, eight parallel pi-stacking interactions with Tyr-185 were identified across seven representative models.

From the PLIP analysis of our docking study, we find of most importance skatole and indole not forming contacts with Ala-73 in CquiOR10 models. By contrast, skatole formed hydrophobic contacts with Leu-73 in 5 of the 10 representative models, while indole formed contacts with Leu-73 in two representative models. This suggests that Ala-73 may indirectly affect specificity by modulating the volume of the binding pocket.

## Appendix 4

Considering both experimental and modeling data, the reason for differing receptor responses at residue 73 could be due to the repacking of this position to accommodate more carbon sidechain atoms that are reducing the binding pocket volume. Leu-73 contains three additional carbon sidechain atoms: γ-carbon, δ1-carbon, and δ2-carbon. Adding these atoms increases the sidechain rotatable bond count from 0 (for Ala-73)–2 (for Leu-73), causing an increase in repackable states for the binding pocket. From our models, the distance of Leu-73 α-carbon to its δ1-carbon or δ2-carbon is approximately 2.8–3.9 Å. The Ala-73 α-carbon to its β-carbon is approximately 1.5 Å. Since the Ala-73 α-carbon is approximately 0.9–1.1 Å inward relative to Leu-73 α-carbon, Leu-73 C-δ atoms could be further inward by approximately 0.2–1.5 Å relative to Ala-73 C-β. This could cause repacking of surrounding sidechains while reducing the volume of the binding pocket, or shift key, nearby residues required for receptor response. Further, from our models, the carbon–carbon length of skatole's methyl group is approximately 1.5 Å. This distance discrepancy, the fact that indole is a rigid molecule, and that the methyl group of skatole is the only rotamer, could also suggest that the binding pocket around position 73 is tightly regulated. The 1.5 Å increase from skatole's methyl group could prevent skatole from occupying the appropriate configuration for receptor response with Leu-73. Conversely, the additional inward 0.2–1.5 Å increase from Leu-73's sidechain could provide indole more contacts or a more favorable configuration with surrounding CquiOR10 sidechains to prompt increased receptor response. Combined, our results suggest that the residue at 73 provide a finely tuned volumetric space to accommodate specific oviposition attractants.

