## [Editor Report]

This article addresses the mechanism of ligand specificity of odorant receptors (OR) through mutational analyses and structure prediction. Through solid data, the authors identify a single amino acid substitution that switches ligand specificity between two olfactory receptors. Obtaining structures of OR complexes has been challenging, so such an approach is valuable and will be of interest to scientists within the fields of chemical ecology and sensory neuroscience.

---

## [Decision Letter]

**Decision letter after peer review:**

Thank you for submitting your article "Two mosquito odorant receptors with reciprocal specificity mediated by a single amino acid residue" for consideration by *eLife*. Your article has been reviewed by 3 peer reviewers, including Sonia Sen as the Reviewing Editor and Reviewer #1, and the evaluation has been overseen by Claude Desplan as the Senior Editor. The following individual involved in review of your submission have agreed to reveal their identity: Kutti R Vinothkumar (Reviewer #2).

Essential revisions:

All three reviewers appreciated the quality and rigour of the work. There were a few points that were raised that would help improve the readability and clarity of the manuscript that are listed below:

1. Writing: While generally clearly written, attention to some aspects of the text will be helpful.

a. Please place the odours in the context of the animal's behaviours in this species and others.

b. At times, the results are difficult to interpret. It would help if the authors could walk the reader through the data before stating their interpretations.

c. Expand explanations such as workflow and logic of selection criteria for which OR chimeras were of interest.

d. The details in the Results section related to the structure are distracting. Could the authors please slim this down, for example, by summarising the interactions with ligand in a tabular form

e. The authors should develop the discussion that currently lacks depth. For example, while the functional analyses are done with the heteromers of OR and ORCO, the structure prediction was done as a homotetramer – some discussion f OR/ORCO homo/hetero tetramer will be valuable. Discuss how their predicted model compares with the known structure and what the authors mean by the contribution of such mechanisms to the evolution of behaviours.

2. Figure 3 and 4 should be merged. In addition, the authors should include the response of the mutant OR2 from current figure 4?

3. The authors should include a supplementary figure that compares the predicted OR10 with the experimentally determined MhOR5. While doing so, they should also discuss how eugenol interacts with the Ala73 residue equivalent in MhOR5?

4. The modelling data appear to be over-interpreted. In light of the functional data for the mutants, their interpretations of the structures of the WT and Ala72Leu mutant might hold true, but should bet toned down and this should include a discussion on the error of the prediction methods.

5. Conservation of the two ORs, particularly the ligand binding pocket: The central claim of the manuscript is that a single residue contributes to ligand specificity. In light of this, it would be nice to talk about the general similarity between the two ORs, particularly within the ligand binding pockets, in the main text along with figure 1 – —figure supplement 1.

In addition to this, do please pay attention to the detailed reviews below.

*Reviewer #1 (Recommendations for the authors):*

We appreciated the manuscript for the clarity of data and writing. We have only a few comments for the authors:

– The manuscript is well written overall, however, the introduction can be improved by including some information on the odors Indole and skatole and their role as oviposition attractants in mosquitoes. How different are these odors behaviorally? The authors can also elaborate more on the said ORs, their function and significance in mosquito behaviors. The citations of the studies which have shown the reverse specificities of OR10 and OR2 are also missing in the introduction. Additionally, a paragraph highlighting the significance of this study can also be included in the introduction/discussion.

– We found the supplementary figures for figures 1 and 2 very useful and recommend including them in the main figure. Particularly Figure 1-supp 1 and 3 and both the figure 2 supplement.

– Currently, figure 2 jumps ahead of the text. This is clarified as one reads on, but could the authors please introduce the work-flow, and what they mean by inner, mid, and outer, before presenting the data?

– It would help to walk the reader through the interpretation of the domain swap experiments in figure 2 and figure 1- supp 2.

– Currently, the authors directly state the inference. For example, from these figures it was not clear to us that TM2 stood out as the candidate. TM2 and TM7 together did, TM7 alone did not, so, was it by mere exclusion at this stage that the authors picked TM2? The later experiments make this clear, of course, but it would be nice to have a more descriptive text for this section.

– Line 119-120 "implications for the evolution of other insect behaviors". This is an interesting point and one worth elaborating on. Line 130: "Optimized the workflow to reduce the number of…" Can the authors please elaborate on how they optimized the workflow? This will be a very useful piece of information for anyone planning to take on a study of this nature.

*Reviewer #2 (Recommendations for the authors):*

General comments

As the analysis of the CquiORs with ligands is derived from models and not experimental structures (in particular they are treated as homotetramers), some of the interactions with ligand might require further study. There is detailed or too much description on how the ligand interacts with the receptor and this is difficult to follow. I have found the manuscript difficult to read and some effort can be made to make it simpler. The discussion is short and could be elaborated.

Specific comments

1) The title – would it help to change "Single amino acid residue mediates reciprocal specificity in two mosquito odorant receptors"

2) In the abstract, line 22, "swapped the transmembrane (TM) domains" is perhaps more appropriate than 'swapped the seven transmembrane (TM) domains' as the first TMD (alone) showed no response.

3) The percent similarity/identity of CquiOR10 and CquiOR2 in the TMDs can be mentioned in the introduction. This will be useful in highlighting how these receptors are able to distinguish a methyl group (though the inner part of TM2 is very similar).

4) It is mentioned that none of the S1 chimeric version (TMD1) showed no response. Has the expression of the chimera been tested i.e., protein in oocytes

5) One supplementary figure of OR10 (from RoseTTAfold or Alphafold) with and overlay of MhraOR5 will be useful. I think these algorithms are very good but an overall picture will be a good addition.

6) In figures (figure 6 -supplementary figures of the main figures 5, 6 – the TMD are shown in gray, which is very clear but will be useful to have the TMD's labelled in the panels.

7) What would be the equivalent residue of Ala73 (CquiOR10) and L74 (CquiOR2) in MhraOR5. Does this residue interact with Eugenol or remains like Ala73 in CquiOR10 (no interaction to ligand)

8) Figure 7 (main figure) would look better, if it has more clarity (looks like it is shaded or some transparency is added).

9) Figure 7 – supplement figure 1, has OLC12 mentioned in the figure (and OCL12 in legend in probably a typo). What is OLC12 and it is mentioned in main text (line 425), is there any significance (it is indicated in acknowledgements as ORCO ligand candidate)

10) Do the CquiOR10 and CquiOR2 form homotetramers or heteromers with ORCO. As the functional analysis of the chimera and mutants done with co-expression, a mention of this and implications if any of heteromer in odorant sensing can be included in the discussion.

11) Was the Rosettaligand docking performed on monomer or homotetramer (models derived from the RoseTTAfold).

*Reviewer #3 (Recommendations for the authors):*

The conclusions of the paper are mostly well-supported by the data, but the authors often overinterpret the modelling results.

1) The data in Figure 4 nicely illustrates the effect of the single amino acid substitution switching the ligand-binding profile. But, CquiOr10 A73L is conspicuously absent from these data and should also be tested for responses to methylindoles.

2) While I believe that the general conclusions are correct, the authors often interpret the modelling results as if they were high-resolution structures. For example, Figure 7 highlights a 1 A deviation between CquiOr10 and the Ala73Leu mutant. But the authors note that the RosettFold and AlphaFold models of CquiOr10 have a helical rmsd of 1.7 A, which suggests to me that structural deviations on this scale should be considered "within error" of the modelling process. Indeed, in Figure 7, several other helices that are far from Ala73 also seem to be in slightly different positions, further suggesting these small deviations may be artifacts of the structure prediction software. I would prefer the authors restrict their interpretation of the modelling and docking results to broad themes (i.e. bigger residue has less space for bulkier ligands) and use them to generate hypotheses for subsequent empirical tests.

The description of docking results is unnecessarily long. For example, lines 351-360 seem to outline the basic conclusions but then lines 362-406 contain many lines listing specific interactions that may happen only rarely – these interactions could be summarized better.

Similarly, long descriptions of structural changes are also not needed. For example, in the Discussion, lines 479-492 describe the additional carbons present in Leu compared to Ala, that these carbons have more rotatable bonds, and that there are a few angstroms differences in positions based on their modelling studies – all to conclude that Leu is bigger than Ala.

It seems that the binding pockets are likely to be highly similar between CquiOr10 and CquiOr2. For example, in Figure 6 the authors show CquiOr10 residues 69, 72, 185, 261, and 294 along with Ala/Leu 73. All of these residues except one (residue 69) are conserved in CquiOr2. This conservation adds strength to the author's argument that the ligand-binding specificity could be determined by a single amino acid residue since much of the ligand-binding surface might be identical. The authors should look more deeply at the conservation within the ligand-binding pockets.

---

## [Author Response]

Essential revisions:All three reviewers appreciated the quality and rigour of the work. There were a few points that were raised that would help improve the readability and clarity of the manuscript that are listed below:1. Writing: While generally clearly written, attention to some aspects of the text will be helpful.a. Please place the odours in the context of the animal's behaviours in this species and others.

In the revised version of the manuscript, we describe the ecological significance of these odorants vis-à-vis insect (mosquito) olfaction.

b. At times, the results are difficult to interpret. It would help if the authors could walk the reader through the data before stating their interpretations.c. Expand explanations such as workflow and logic of selection criteria for which OR chimeras were of interest.

We have followed these suggestions closely. Moving child figures to main figures (per the suggestion below) helped us set the stage before describing the results.

d. The details in the Results section related to the structure are distracting. Could the authors please slim this down, for example, by summarising the interactions with ligand in a tabular form

We have summarized the Results section related to structural details and have included a detailed Appendix. The interactions calculated from PLIP are summarized in supplemental tables 5-13.

e. The authors should develop the discussion that currently lacks depth. For example, while the functional analyses are done with the heteromers of OR and ORCO, the structure prediction was done as a homotetramer – some discussion f OR/ORCO homo/hetero tetramer will be valuable. Discuss how their predicted model compares with the known structure and what the authors mean by the contribution of such mechanisms to the evolution of behaviours.

We have followed these suggestions closely and provided an in-depth discussion of the current results. We discussed OR homomers vs. heteromers and put the results in the context of the field's current status. However, we omitted the evolutionary context statement, which might distract from the research's primary focus.

2. Figure 3 and 4 should be merged. In addition, the authors should include the response of the mutant OR2 from current figure 4?

We merged these figures as recommended.

3. The authors should include a supplementary figure that compares the predicted OR10 with the experimentally determined MhOR5. While doing so, they should also discuss how eugenol interacts with the Ala73 residue equivalent in MhOR5?

We have created a new figure that shows the MhOR5 with CquiOR10 RoseTTAFold and AlphaFold models. The Results section highlights a MhOR5 TM4 sequence motif interacting with eugenol similar to CquiOR10 TM2 from Needleman-Wunsch pairwise alignment. Notably, this sequence motif also contains CquiOR10Ala73. In our modeling, CquiOR10Asn72 is a skatole/indole interacting residue matching the eugenol-interacting MhOR5Ile213.

4. The modelling data appear to be over-interpreted. In light of the functional data for the mutants, their interpretations of the structures of the WT and Ala72Leu mutant might hold true, but should bet toned down and this should include a discussion on the error of the prediction methods.

We have trimmed the Discussion section to provide more generalized conclusions and added an Appendix to describe our interpretation of our proposed space-constrained hypothesis in detail. We have also included more detail on the error of these modern modeling approaches and have emphasized the challenges with these proteins given the lack of closely homologous structures. We have also included a sentence describing how future experimental structures can help us test the hypotheses presented in our work.

5. Conservation of the two ORs, particularly the ligand binding pocket: The central claim of the manuscript is that a single residue contributes to ligand specificity. In light of this, it would be nice to talk about the general similarity between the two ORs, particularly within the ligand binding pockets, in the main text along with figure 1 – —figure supplement 1.

We have generated an additional figure showing the Needleman-Wunsch pairwise alignment between CquiOR10 and CquiOR2, with 49.5% sequence identity (189/382 residues) and 71.7% sequence similarity (274/382 residues). Further, we have included an additional table highlighting from our modeling studies that the CquiOR10 residues interacting with skatole/indole from RosettaLigand/PLIP were physiochemically identical or similar to CquiOR2; this has also been mentioned in the Results section.

Reviewer #1 (Recommendations for the authors):We appreciated the manuscript for the clarity of data and writing. We have only a few comments for the authors:– The manuscript is well written overall, however, the introduction can be improved by including some information on the odors Indole and skatole and their role as oviposition attractants in mosquitoes. How different are these odors behaviorally? The authors can also elaborate more on the said ORs, their function and significance in mosquito behaviors. The citations of the studies which have shown the reverse specificities of OR10 and OR2 are also missing in the introduction. Additionally, a paragraph highlighting the significance of this study can also be included in the introduction/discussion.

We have followed these constructive suggestions very closely.

– We found the supplementary figures for figures 1 and 2 very useful and recommend including them in the main figure. Particularly Figure 1-supp 1 and 3 and both the figure 2 supplement.

We followed your recommendation, and they help the reader follow the flow of the work more efficiently.

– Currently, figure 2 jumps ahead of the text. This is clarified as one reads on, but could the authors please introduce the work-flow, and what they mean by inner, mid, and outer, before presenting the data?– It would help to walk the reader through the interpretation of the domain swap experiments in figure 2 and figure 1- supp 2.

Bringing the child figures into the primary text helped us address these legitimate concerns.

– Currently, the authors directly state the inference. For example, from these figures it was not clear to us that TM2 stood out as the candidate. TM2 and TM7 together did, TM7 alone did not, so, was it by mere exclusion at this stage that the authors picked TM2? The later experiments make this clear, of course, but it would be nice to have a more descriptive text for this section.

Indeed, it was intriguing that TM2 alone did not work, but given the volume of the work, we did not go back and try to address the issue, which became a moot point. We decided to move on once we identified the critical mutant M2,M7.

– Line 119-120 "implications for the evolution of other insect behaviors". This is an interesting point and one worth elaborating on. Line 130: "Optimized the workflow to reduce the number of…" Can the authors please elaborate on how they optimized the workflow? This will be a very useful piece of information for anyone planning to take on a study of this nature.

We describe how we optimize the workflow to reduce the number of multiple-point mutations. However, we omitted the statement in line 119 to avoid distracting the reader from the main findings of the work.

Reviewer #2 (Recommendations for the authors):General commentsAs the analysis of the CquiORs with ligands is derived from models and not experimental structures (in particular they are treated as homotetramers), some of the interactions with ligand might require further study. There is detailed or too much description on how the ligand interacts with the receptor and this is difficult to follow. I have found the manuscript difficult to read and some effort can be made to make it simpler. The discussion is short and could be elaborated.

These concerns have been addressed, and the Discussion has been expanded.

Specific comments1) The title – would it help to change "Single amino acid residue mediates reciprocal specificity in two mosquito odorant receptors"

We changed the title as suggested.

2) In the abstract, line 22, "swapped the transmembrane (TM) domains" is perhaps more appropriate than 'swapped the seven transmembrane (TM) domains' as the first TMD (alone) showed no response.

We have deleted the word "seven."

3) The percent similarity/identity of CquiOR10 and CquiOR2 in the TMDs can be mentioned in the introduction. This will be useful in highlighting how these receptors are able to distinguish a methyl group (though the inner part of TM2 is very similar).

We appreciate this suggestion. The introduction now includes the Needleman-Wunsch pairwise identity and similarity of the entire sequence and the ranges TM1-TM6 sequence identity, with a supplemental table providing further detail.

4) It is mentioned that none of the S1 chimeric version (TMD1) showed no response. Has the expression of the chimera been tested i.e., protein in oocytes

No, we did not check protein expression. We used the Orco ligand candidate OLC12 as an indicator. Our laboratory and others observed that the responses to Orco agonists, like OLC12, are more robust when heteromers (OR-Orco) are formed than homomers (Orco-Orco). We provided various references to the literature in the revised version of the manuscript.

5) One supplementary figure of OR10 (from RoseTTAfold or Alphafold) with and overlay of MhraOR5 will be useful. I think these algorithms are very good but an overall picture will be a good addition.

We have created a new figure that shows the MhOR5 overlaid with CquiOR10 RoseTTAFold and AlphaFold models

6) In figures (figure 6 -supplementary figures of the main figures 5, 6 – the TMD are shown in gray, which is very clear but will be useful to have the TMD's labelled in the panels.

We have revised the main figures.

7) What would be the equivalent residue of Ala73 (CquiOR10) and L74 (CquiOR2) in MhraOR5. Does this residue interact with Eugenol or remains like Ala73 in CquiOR10 (no interaction to ligand)

In the Results section, we highlighted a MhOR5 TM4 sequence motif interacting with eugenol similar to CquiOR10 TM2 from Needleman-Wunsch pairwise alignment. Notably, this sequence motif also contains CquiOR10Ala73. In our modeling, CquiOR10Asn72 is a skatole/indole interacting residue matching the eugenol-interacting MhOR5Ile213. In the MhOR5-eugenol structure (PDB ID: 7LID), the CquiOR10Ala73 equivalent residue is Thr214; from PLIP, this residue does not interact with eugenol

8) Figure 7 (main figure) would look better, if it has more clarity (looks like it is shaded or some transparency is added).

Indeed. As suggested, a revision of the figure brought more clarity to the reader.

9) Figure 7 – supplement figure 1, has OLC12 mentioned in the figure (and OCL12 in legend in probably a typo). What is OLC12 and it is mentioned in main text (line 425), is there any significance (it is indicated in acknowledgements as ORCO ligand candidate)

Thank you for pointing it out. We fixed this oversight. In the main text, we described OLC12 and provided relevant references.

10) Do the CquiOR10 and CquiOR2 form homotetramers or heteromers with ORCO. As the functional analysis of the chimera and mutants done with co-expression, a mention of this and implications if any of heteromer in odorant sensing can be included in the discussion.11) Was the Rosettaligand docking performed on monomer or homotetramer (models derived from the RoseTTAfold).

These relevant issues #11 and #12 have been addressed in the second paragraph in the Discussion (line 556).

Reviewer #3 (Recommendations for the authors):The conclusions of the paper are mostly well-supported by the data, but the authors often overinterpret the modelling results.1) The data in Figure 4 nicely illustrates the effect of the single amino acid substitution switching the ligand-binding profile. But, CquiOr10 A73L is conspicuously absent from these data and should also be tested for responses to methylindoles.

We debated at some point whether it would be necessary to test these “less potent” ligands, given the crystal clear picture generated with the most potent ligands (skatole and indole). Regarding the methylindoles, in particular, it is worth noting that CquiOR2 does not discriminate 1, 2, 4, 5, 6, and 7-methylindole. Thus, it is unlikely that the responses elicited by CquiOR10A73L would add relevant information. By contrast, CquiOR10 elicited larger currents when stimulated with two ligands (1- and 5-methylindole). Thus, we interrogated whether CquiOR274A would recapitulate that response profile.

2) While I believe that the general conclusions are correct, the authors often interpret the modelling results as if they were high-resolution structures. For example, Figure 7 highlights a 1 A deviation between CquiOr10 and the Ala73Leu mutant. But the authors note that the RosettFold and AlphaFold models of CquiOr10 have a helical rmsd of 1.7 A, which suggests to me that structural deviations on this scale should be considered "within error" of the modelling process. Indeed, in Figure 7, several other helices that are far from Ala73 also seem to be in slightly different positions, further suggesting these small deviations may be artifacts of the structure prediction software. I would prefer the authors restrict their interpretation of the modelling and docking results to broad themes (i.e. bigger residue has less space for bulkier ligands) and use them to generate hypotheses for subsequent empirical tests.

We have summarized the Results section related to structural details and have included an Appendix with additional details. We have trimmed the Discussion section to provide more generalized conclusions and added a supplemental text to describe our interpretation of our proposed space-constrained hypothesis in detail. We have also included more detail on the error of these modern modeling approaches and have emphasized the challenges with these proteins, given the lack of closely homologous structures. We have also included a sentence describing how future experimental structures can help us test the hypotheses presented in our work.

The description of docking results is unnecessarily long. For example, lines 351-360 seem to outline the basic conclusions but then lines 362-406 contain many lines listing specific interactions that may happen only rarely – these interactions could be summarized better.

We have summarized sections related to structural details and have included a supplemental text with additional detail. The interactions calculated from PLIP are summarized in supplementary tables 5-13.

Similarly, long descriptions of structural changes are also not needed. For example, in the Discussion, lines 479-492 describe the additional carbons present in Leu compared to Ala, that these carbons have more rotatable bonds, and that there are a few angstroms differences in positions based on their modelling studies – all to conclude that Leu is bigger than Ala.

This section emphasizes how the receptor’s volumetric space could be constrained by more than the distance of an additional methyl group. Rather than concluding that leucine is larger than alanine, we are trying to convey that CquiOR10 Leu-73 could prevent skatole from occupying the appropriate configuration compared to indole. We have trimmed this section in hopes of clarifying this hypothesis. The additional details outlining how we reached this hypothesis are in the supplemental text.

It seems that the binding pockets are likely to be highly similar between CquiOr10 and CquiOr2. For example, in Figure 6 the authors show CquiOr10 residues 69, 72, 185, 261, and 294 along with Ala/Leu 73. All of these residues except one (residue 69) are conserved in CquiOr2. This conservation adds strength to the author's argument that the ligand-binding specificity could be determined by a single amino acid residue since much of the ligand-binding surface might be identical. The authors should look more deeply at the conservation within the ligand-binding pockets.

We have generated an additional figure showing the Needleman-Wunsch pairwise alignment between CquiOR10 and CquiOR2, with 49.5% sequence identity (189/382 residues) and 71.7% sequence similarity (274/382 residues). Further, we have included an additional table highlighting from our modeling studies that the CquiOR10 residues interacting with skatole/indole from RosettaLigand/PLIP were physiochemically identical or similar to CquiOR2; this has also been mentioned in the Results section. We have added a supplemental table with Needleman-Wunsch pairwise identity and similarity of t ranges TM1-TM6 from OCTOPUS transmembrane prediction.